# PopSign ASL v1.0: An Isolated American Sign Language Dataset Collected via Smartphones

**Thad Starner**[1]  **Sean Forbes**[2]  **Matthew So**[1]  **David Martin**[1]  **Rohit Sridhar**[1]
**Gururaj Deshpande**[1]  **Sam Sepah**[3]  **Sahir Shahryar**[1]  **Khushi Bhardwaj**[1]
**Tyler Kwok**[1]  **Daksh Sehgal**[1]  **Saad Hassan**[4]  **Bill Neubauer**[1]
**Sofia Anandi Vempala**[1]  **Alec Tan**[1]  **Jocelyn Heath**[1]  **Unnathi Utpal Kumar**[1]
**Priyanka Vijayaraghavan Mosur**[1]  **Tavenner M. Hall**[1]  **Rajandeep Singh**[1]
**Christopher Zhang Cui**[1]  **Glenn Cameron**[3]  **Sohier Dane**[3]  **Garrett Tanzer**[3]

[1]Georgia Institute of Technology   [2]Deaf Professional Arts Network
[3]Google   [4]Rochester Institute of Technology

{thad,matthew.so,dmartin305,rkohitsridhar,gurudesh}@gatech.edu
{khushi.bhardwaj,tkwok7,dsehgal34,wneubauer4}@gatech.edu
{svempala7,atan70,jheath38,unnathikumar}@gatech.edu
{pmosur3,rajandeep.singh,ccui46}@gatech.edu
sean@dpan.tv, sh253@rit.edu, sahirshahryar@gmail.com
tavennerh@yahoo.com, {sepah, glencam, sohier, gtanzer}@google.com

## Abstract

PopSign is a smartphone-based bubble-shooter game that helps hearing parents of deaf infants learn sign language. To help parents practice their ability to sign, PopSign is integrating sign language recognition as part of its gameplay. For training the recognizer, we introduce the PopSign ASL v1.0 dataset that collects examples of 250 isolated American Sign Language (ASL) signs using Pixel 4A smartphone selfie cameras in a variety of environments. It is the largest publicly available, isolated sign dataset by number of examples and is the first dataset to focus on one-handed, smartphone signs. We collected over 210,000 examples at 1944x2592 resolution made by 47 consenting Deaf adult signers for whom American Sign Language is their primary language. We manually reviewed 217,866 of these examples, of which 175,022 (approximately 700 per sign) were the sign intended for the educational game. 39,304 examples were recognizable as a sign but were not the desired variant or were a different sign. We provide a training set of 31 signers, a validation set of eight signers, and a test set of eight signers. A baseline LSTM model for the 250-sign vocabulary achieves 82.1% accuracy (81.9% class-weighted F1 score) on the validation set and 84.2% (83.9% class-weighted F1 score) on the test set. Gameplay suggests that accuracy will be sufficient for creating educational games involving sign language recognition.

## 1   Introduction

PopSign is a bubble-shooter smartphone-based game that helps hearing parents of deaf infants learn American Sign Language. It builds on significant past user studies on such games [Xu, 2013, Summet, 2010]. PopSign focuses on vocabulary from the MacArthur-Bates Communicative Development Inventories (CDI) [Fenson and Marchman, 2007], which are the first concepts one teaches a child in any language. Originally, PopSign required players to recognize a short video of a sign and match it to one of five types of bubbles labeled with English words. The player shoots a bubble and, if it strikes two or more contiguous bubbles of the same type, all the bubbles disappear. The goal is to clear the screen. The initial game was enjoyable but had limitations as it solely emphasized receptive

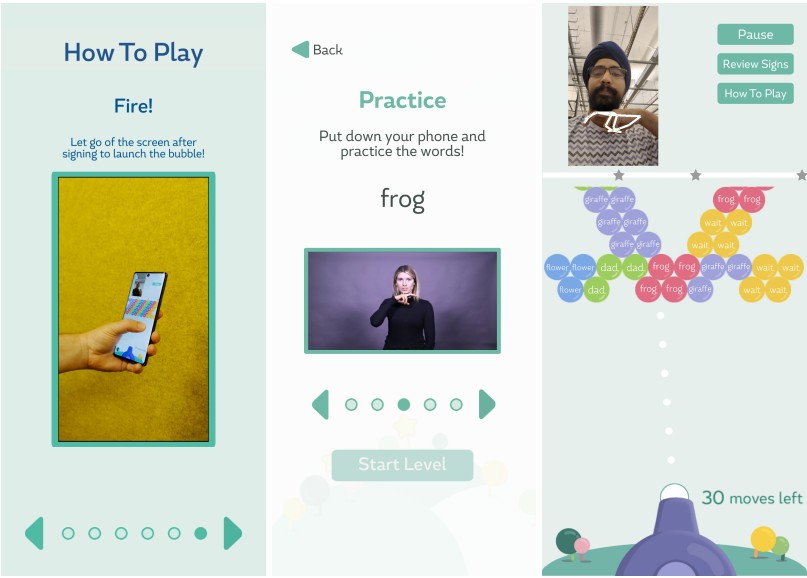

Figure 1: PopSignAI incorporates sign recognition as part of PopSign's bubble shooter game play. Left: Instructions. Middle: Practicing five signs before each level. Right: The player holds their phone and touches the screen with their thumb to aim the bubble. The other hand performs the sign to select which type of bubble will be shot. Here, the player is aiming at the red balls and signing FROG. Google's MediaPipe hand tracking overlays the player's selfie camera video in real-time.

skills. Incorporating sign language recognition into the game (see Figure 1) allows parents to practice signing themselves, developing expressive rather than just receptive signing skills. These expressive skills are necessary in order for hearing parents to teach sign language to their infants.

To create a version of PopSign that includes sign recognition, we collected a large number of examples of 250 individual signs from people for whom ASL is their primary language. Since PopSign only engages five English/ASL concepts at a time, the expectation is that a recognition engine, coupled with a hand tracker such as Google's MediaPipe, could run sufficiently well on a smartphone to support gameplay [Lugaresi et al., 2019]. This assumption has proven correct. The eventual goal is a general-purpose recognition engine developed for the full 250-sign vocabulary as a plug-in for popular game creation engines, such as Unity, to support development of educational games in sign.

## 2   Background and Motivation

Ninety-five percent of deaf children are born to hearing parents [Moores, 2000], and most of those parents never learn enough sign language to teach their children. The majority of deaf children of hearing parents remain significantly delayed in language development throughout their lives when compared with hearing children and deaf children of deaf parents [Johnson et al., 1989, Spencer, 1993a,b]. The children's slow development of language has been attributed to incomplete language models and interaction [Hamilton and Lillo-Martin, 1986, Lederberg and Everhart, 1998]. An environment without access to language results in language deprivation, which leads to health, education and quality-of-life issues such as social isolation [Crowe, 2003, Twersky Glasner and Miller, 2010], mental health problems (2-7x that of hearing children) [Dammeyer, 2009], and suicide (6-60x hearing) [Turner et al., 2007, You, 2017]. In many cases, deaf children of hearing parents (in the United States) are fluent neither in English nor ASL [Woll, 1998, Kannapell, 1989]. For these deaf individuals, language is sometimes a life-long struggle [Jean F. Andrews, 2019]. Language acquisition is dependent upon the availability of that language and the opportunities a child [Clark et al., 2001] or an adult learner [Kramsch, 2000, Schutz, 2005] have for interacting with skilled users of the language. Deaf children of hearing parents typically grow up in linguistically impoverished surroundings due to the inability of family members to use sign [Goldin-Meadow, 1999, Goldin-Meadow and Mylander, 1990]. The quantity and quality of adult-child language interaction at an

early age has also been shown to affect the language development and subsequent school success of hearing children [Panagos, 1998].

Once a decision is made to use sign language, which 75% of parents eventually do [Meadow-Orlans et al., 2003], the family faces the task of learning a new language in a modality (visual-gestural) that is foreign to them as the main avenue of communication. Local ASL classes may not focus on the topics and vocabulary that new parents need; one-on-one family government services may be helpful but are often infrequent; classes may not be accessible in terms of cost or location; and books can be dry, demotivating, and not the best method for conveying ASL [Summet, 2010]. Given these daunting obstacles, many families may opt to follow the advice of doctors and adopt cochlear implant and spoken language-only approaches Hall [2017]. However, it has been demonstrated that not only is some level of ASL exposure for deaf children empirically not harmful to their development Hall et al. [2019] and vastly superior to no language exposure [Singleton and Newport, 2005], but in fact, sign language is more effective at reducing cognitive delays and improving communication skills than hearing-based approaches Hall [2017], particularly among the many children whose brains are unable to understand speech through cochlear implants Humphries et al. [2012]. Further, while vocabulary knowledge alone does not constitute fluency or command of ASL, it has been shown that a knowledge of approximately 150 words is enough to increase the rate at which new words and grammatical skills are acquired [Marchman and Bates, 1994]. Even "survival level" signing is a worthwhile endeavor for families with deaf children Berger et al. [2023]. PopSign seeks to help address this silent crisis by providing hearing parents a tool by which they may learn ASL vocabulary.

## 2.1 Isolated Sign Language Recognition (ISLR) Datasets

PopSign ASL v1.0 collects examples of 250 isolated American Sign Language signs using the selfie camera on Pixel 4A smartphones in a variety of environments. It is the largest isolated sign language dataset publicly available, the first to focus on one-handed signing with smartphones, and one of the few of its size that has been manually reviewed.

The majority of isolated sign datasets are collected in laboratory settings with controlled lighting and angles. Some have been collected by users with their own cameras in private settings. Sign language data collection carries its own unique challenges. For one, the data collected is usually in video format [Quer and Steinbach, 2019]. Video introduces variables such as lighting, background textures, field of view, resolution, participant diversity, and cost. Since participants are directly recorded, there are additional challenges in acquiring consent [Quer and Steinbach, 2019]. Finally, ASL has a vocabulary with many thousands of signs and varies considerably by region, ethnicity, gender, and even by household [LeMaster and Monaghan, 2005].

Table 1 details major publicly available ISLR datasets. ASLLVD, Purdue86RVL-SLLL, and RWTH-BOSTON 50 were all collected in a studio setting [Athitsos et al., 2008, Martinez et al., 2002, Zahedi et al., 2005]. More recent efforts include WLASL, ASL-100-RGBD, MS-ASL, and ASL Citizen [LI et al., 2020, Joze and Koller, 2018, Desai et al., 2023]. ASLLVD features a large number of signs, but the dataset has few videos per sign. Furthermore, since there are six signers at most per sign, generalization across users is difficult. RWTH-Boston 50 and WLASL suffer from the same problem of having few videos per sign but not to the same degree. Purdue RVL-SLLL addresses the issue of having too few videos per sign, but it does not feature many signers. All of those datasets have been collected using studios for recording, which is too prohibitively expensive for scalable data collection. ASL Citizen fixes many of these issues by collecting a very large number of videos over many signs and features enough participants to increase generalizability over users. The signs are collected via webcam, making large scale data collection less expensive. MS-ASL is cost-effective in collection, since it is scraped from public videos. However, participants do not explicitly consent to such scraped collections, and annotation can be difficult (as highlighted by Bragg et al. [2021].) There are many datasets for other sign languages, some of which are listed in Table 1. We note that these datasets suffer from the same issues in existing ASL datasets. In particular, there are often too few signs and signers or too few examples per sign for useful downstream applications. Furthermore, these sets are recorded under tightly controlled studio setups, limiting data collection and utility.

| Dataset | Lang | Signs | Signers | Videos | Videos/Sign | Source | Citation |
|---|---|---|---|---|---|---|---|
| ASLLVD | ASL | 3000 | 6 | 9800 | 3.27 | Studio | 2008 |
| Purdue RVL-SLLL | ASL | 101 | 14 | 2576 | 25.50 | Studio | 2002 |
| RWTH-BOSTON 50 | ASL | 50 | 3 | 483 | 9.66 | Studio | 2005 |
| WLASL | ASL | 2000 | 119 | 21083 | 10.54 | Studio | 2020 |
| MS-ASL | ASL | 1000 | 222 | 25513 | 25.51 | Web | 2018 |
| ASL Citizen | ASL | 2731 | 52 | 83912 | 30.73 | Webcam | 2023 |
| **PopSign ASL v1.0** | **ASL** | **250** | **47** | **214326** | **857.30** | **Smartphone** | |
| BSLDict | BSL | 9283 | 148 | 14210 | 1.53 | Studio | 2020 |
| DEVISIGN-G | CSL | 36 | 8 | 432 | 12.00 | Studio | 2013 |
| DEVISIGN-D | CSL | 500 | 8 | 6000 | 12.00 | Studio | 2013 |
| DEVISIGN-L | CSL | 2000 | 8 | 24000 | 12.00 | Studio | 2013 |
| CSL 500 | CSL | 500 | 50 | 125000 | 250.00 | Studio | 2019 |
| DGS Kinect 40 | DGS | 40 | 14 | 2800 | 70.00 | Studio | 2012 |
| SMILE | DSGS | 100 | 30 | - | - | Studio | 2018 |
| GSL 982 | GSL | 982 | 1 | 4910 | 5.00 | Studio | 2012 |
| INCLUDE | ISL | 263 | 7 | 4287 | 16.30 | Studio | 2020 |
| LSA64 | LSA | 64 | 10 | 3200 | 50.00 | Studio | 2016 |
| LSE-Sign | LSE | 2400 | 2 | 2400 | 1.00 | Studio | 2015 |
| LSFB-ISOL | LSFB | 395 | 100 | 47551 | 120.38 | Studio | 2021 |
| BosphorusSign | TSL | 855 | 10 | >51300 | >60.00 | Studio | 2016 |
| BosphorusSign22K | TSL | 744 | 6 | 22542 | 30.30 | Studio | 2020 |
| AUTSL | TSL | 226 | 43 | 38336 | 169.63 | Studio | 2020 |

Table 1: Isolated Sign Language Datasets

## 2.2 Smartphone Signing

We took an approach similar to ASL Citizen but used smartphone selfie cameras for data collection. Smartphone cameras tend to be of higher quality than webcams or cameras embedded in a laptop. Laptop and webcam cameras often have a smaller field-of-view than recent smartphones, making it difficult for a signer to interact with a keyboard or mouse during data collection and still be distant enough to capture the full signing box (the volume of space in which signers move their hands while signing). Our ASLRecorder data collection smartphone app (see below) requires the user to hold and interact with the phone with one hand while signing with the other. This posture is surprisingly commonplace for Deaf signers, who often make video calls from their smartphones. Signers either hold the phone or rest it on a desk or their body while communicating. Signers are used to aligning the camera so that the lighting is sufficient and the camera can see the signing hand and the face. While many signs are two-handed, smartphone signing Morris [2022b] has become so common that many signers will adjust their signs so that they can be made with one hand or fingerspell a sign if the one-handed version is ambiguous. From a collection standpoint, smartphones are simple to ship to participants, the collection system is self-contained, and most participants are familiar with how to use a smartphone.

## 3 Collection Methodology

The Deaf Professional Arts Network (DPAN) recruited 50 Deaf signers for whom ASL was their primary language, and 47 completed the task. DPAN shipped Pixel 4A smartphones to each participant with the ASLRecorder open-source data collection smartphone app[1], and participants were given guidance on using it[2]. Participants returned the phones after completion. Upon receipt, DPAN confirmed that all recorded videos had been uploaded to an on-line repository.

## 3.1 Choosing Lemmas

Examples of processes for choosing which lemmas to include in a sign language dataset can be found for British Sign Language Fenlon et al. [2015], Sign Language of the Netherlands (NGT) Schüller

---

[1]https://github.com/matthew-so/ASLRecorder/
[2]https://tinyurl.com/2p99s299

et al. [2021], ASL Hochgesang et al. [2018] and others. The closest for our purposes is the ASL CDI 2.0 Caselli et al. [2020] which was constructed by three expert signers updating the CDI for concepts used with Deaf infants. For selecting the 250 one-handed signs in our subset of the CDI, DPAN created reference videos based on the English version of all 563 concepts in the CDI as used in Xu's research on ASL games Xu [2013], which was taken from Anderson and Reilly [2002]. Five DPAN members, who have signed since childhood, agreed on how to sign each concept one-handed as if they were on a video conference and holding a smartphone with one hand. These five signers come from diverse regions and schools. Example videos were made for each agreed-upon sign. From these 563 videos, 250 signs were chosen in an attempt to avoid ones that are too similar (with limited success, see below). The PopSign game uses these reference videos to teach players how to sign.

## 3.2 ASLRecorder App

The ASLRecorder app was designed to terminate after 200 signs to encourage regular breaks throughout the task. Participants could restart the app to continue. Each signer provided approximately 5000 examples (20 examples of 250 individual signs). Participants were encouraged to sign in varying lighting conditions and with varying backdrops. Participants were discouraged from including others in their videos. The example video of each sign from DPAN could be accessed by the participants by clicking on the icon at the top of the recording screen.

ASLRecorder attempts to capture participants' signing in a manner similar to how the PopSign game is played. PopSign uses a hold-to-aim mechanism where the player signs the name of the intended bubble before releasing the screen; therefore, ASLRecoder implements a hold-to-record mechanism. Because phones cannot immediately begin and stop recording, we decided to record continuously. Saving the press and release events as timestamps allows us to easily adjust the start and end point of each clip later. We stored timestamps in the image description (EXIF) metadata of a thumbnail for each video. In order to manage the file size of a continuous recording, we imposed a fifteen-minute limit on the recording, after which the video stops automatically. Videos are recorded using the selfie camera of the phone at 1944x2592 resolution at 30 frames per second. To improve the reliability of ASLRecorder and reduce infrastructure overhead, we used Google Photos's automatic cloud backup service to store the recordings. Each signer has a separate Google account associated with their phone that provides a backup folder and silos each volunteer's data from that of the other participants. The phone automatically connects to the signer's Wi-Fi network and uploads the videos whenever the phone is powered externally. This feature allows DPAN staff to observe a participant's progress and monitor for any difficulties remotely.

ASLRecorder has 10 randomly-selected signs per session that are presented in a sliding user interface, allowing users to freely go back and forth between the selected signs. The interface automatically moves to the next sign after the user releases the Record button, but users can swipe back to the previous sign if they wish to make a better recording. Because we only ask them to make one recording for each sign in a session, we assume that only the last recording for each sign is valid (within that session). At the end of each session, signers have the opportunity to review their videos and manually mark them as invalid if they felt they had performed the sign improperly. We kept timestamps for invalid clips, but they were not included in the training data.

Since the individual sign videos were recorded as continuous sessions, we decoded the videos and split them into individual videos for each recording using a custom-made script [3]. Decoding strategies generally followed the assumption that, by instruction, users held the record button for the duration of their sign. We added a buffer of 0.5 seconds by default on either side of the recording as users tend to press too late and release to early. Of 47 initial signers recruited, 31 signers could be decoded with this approach, splitting the 200 sign videos (plus redos) into individual videos for review. The recordings of 10 signers had to be split after making some manual adjustments to the buffer size. The remaining six signers tapped the button to start signing, rather than holding the button. Generally, these participants completed each sign in two seconds, allowing segmentation with a modified script.

## 3.3 Review Procedure

We designed a custom annotation engine to review the sign videos, given in Figure 2. We classify our data into three categories:

---

[3] `https://github.com/matthew-so/Mobile-Data-Processing-Pipeline/decode.py`

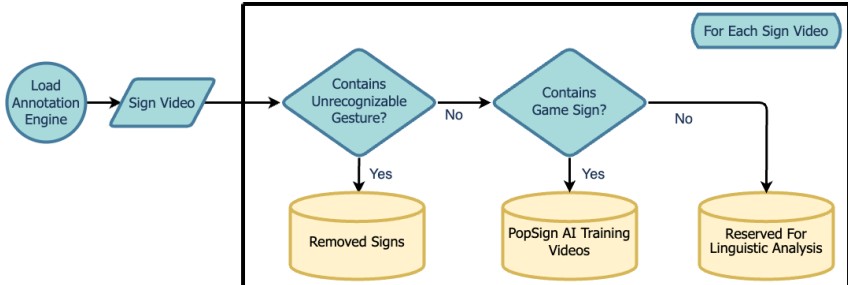

Figure 2: Review process.

- the sign to be taught in the PopSign game
- incorrect signs or unintended variants of the game version of the sign
- unrecognizable videos

A video is marked unrecognizable if the video is unplayable, empty, or does not include the hands or fingers for most of the video. Videos with an identifiable person (i.e., the face was recognizable) besides the participant were also rejected. We mark videos with signs that do not match the DPAN reference sign as a variant. These include signs that are distinct but have the same meaning as the original sign, as well as signs for the same gloss word, but with different meanings. Repetitive motions within signs were accepted as they are common in ASL and understood by a viewer. An example is SHOWER, with the fingers transitioning from closed to spread once, twice or three times.

Since the DPAN reference signs will be used for teaching vocabulary in the PopSign game, we use these videos for judging whether an example should be accepted. Reviewers include 13 undergraduate and graduate students from Georgia Tech, a volunteer, and a professor. The team is supervised by a Deaf researcher who advises on linguistic variants of the signs. Three signers on the team split the vocabulary into 130 signs that were expected to have a low number of variants and 120 that were expected to have a high number of variants. Reviewers with familiarity of ASL were assigned to the signs with greater number of variants. Review was performed one sign at a time using a custom annotation engine [4]. Along with classification, reviewers made notes in a coordination spreadsheet as to the types of variants they observed. Reviewers also provided status reports, bugs, feedback and feature requests to improve the annotation software.

## 4    Dataset Composition

217,865 videos were reviewed. Of these, 3,539 videos were rejected, and 175,022 were considered to be the desired sign, making an average of 700 correct examples per sign. 39,304 videos were judged to be variants of the intended sign or a different sign. As the variants can be informative for creating sign recognition systems for other purposes, we include them in the dataset [5], resulting in a total of 214,326 total videos. PEN has 907 total examples, making it the most represented in the dataset. CAR is the least represented with 467 total examples. Counting the number of examples to be used for training the game, SCISSORS, with 888 examples, is most represented, while SHOWER, with 96, is the least represented. More information can be found in the supplementary material. Of the 47 signers in the dataset, 16 identify as male and 31 as female, and 13 are left-handed and 34 are right-handed. The dataset represents a mix of skin tones and ages. Some participants would switch their signing hand during collection.

There were some homophones and some near-homophones in the dataset. FINGER/WAIT and MOUTH/LIPS are only differentiated by mouth movements in the reference videos. TOOTH/GLASSWINDOW and CHIN/LIPS have minor position differences relative to the lips. WAKE/AWAKE differ by an initial F-hand shape versus A-hand shape. THAT/STAY differ by a wrist rotation. The requirement to sign with one hand resulted in some pairs of signs that normally would be distinct to become very similar. For example, PERSON is performed with two flattened hands,

---

[4]https://github.com/Benler123/hotkey_annotate

[5]http://signdata.cc.gatech.edu

moving in parallel high-to-low on the body. When performed one-handed, the sign can become similar to BESIDE.

The high number of unintended variants in the dataset (roughly 20%) was influenced by using an English gloss for prompting the participants during data collection. Signers may assume they know the sign desired based on the gloss and not refer to the reference example provided. Alternatively, while providing 5000 examples, it is easy to forget which sign was desired, and we observed situations where participants would alternate between two signs. To further complicate matters, some of the English prompts have two meanings which caused some signers at times to provide the sign with the unintended meaning. Examples include CLOSE ("**close** the doors" versus "**close** to me") and DUCK ("swimming **duck**" versus "be sure to **duck** the overhead pipe"). For future data collection, a smaller sized reference video for each sign playing on the collection screen itself may improve consistency. However, even then, common variations which have small distinctions may be overlooked. For example, we requested THERE to be made with an upward facing palm pointing gesture, though it is commonplace to use an index finger pointing gesture similar to how HE/SHE/IT is formed. The distinction could easily be overlooked in the reference video.

On the other hand, prompting with the English gloss has the benefit of collecting more variations of each sign, which could be valuable in making a more general-purpose sign recognition system (though labeling all of these unintended signs is future work). In this dataset, we observed five classes with significant variants: **Compound signs** such as BEDROOM, BACKYARD, and REFRIGERATOR may vary from the reference example by the ordering of the parts or the formation of one or both of the constituent parts. **Lexicalization** results in the parsing of the first letter of a sign's closest gloss and using the manual alphabet as the handshape of the sign. For example, a signer might use a R-hand shape for RED. **Regional and cultural variations** are commonplace. The sign for GARBAGE in this dataset would be understood as CABBAGE in many areas of the United States. PIZZA has a surprising number of regional variations. **Iconic signs** in the vocabulary often had variations. For example, TOOTHBRUSH may be formed by using the index finger as the toothbrush to mimic the back and forth motion. Alternatively, TOOTHBRUSH may be signed by mimicking holding a toothbrush in the palm. VACUUM may be formed by imitating the motion made while vacuuming or by using the hand to mimic the internal fan of the device. **Contextual understanding** of the English gloss also caused variations. BEFORE in time is signed differently than when something comes sequentially BEFORE something else. Other examples include PRETEND, SAME, and CLOSE.

We assign 31 signers to the training set, eight to the validation set, and eight to the test set. The eight chosen for the validation set capture the broadest possible demographic and linguistic range in order to make validation results as representative of real-world performance as possible. Of the eight users, four were noted for exhibiting high variance (differed from DPAN's example sign), while one was noted for exhibiting middle levels of variance. In addition to variability, we classified signers according to the Fitzpatrick scale, a numerical classification of human skin color. Of the validation signers, five had skin phenotype I, two had skin phenotype II, and two had skin phenotype V (a lower-numbered phenotype represents fairer skin). Finally, the gender of the validation users is evenly split between male and female. The test set has demographics proportional to the training set.

# 5    Experiments on PopSign ASL v1.0

Georgia Tech's PACE computing cluster [PACE, 2017] was used to extract features with MediaPipe Hands with a detection confidence of 0.5 and a tracking confidence of 0.1. MediaPipe Hands extracts 21 key points, each containing an x, y and z coordinate [Lugaresi et al., 2019]. Preprocessing includes removing NaNs for each key point. All videos where a left hand was used to sign were flipped along the vertical axis to convert them into right-handed signs. The PopSign game recognizes when a player is signing with the left hand and also flips the axis. Since there are much fewer left hand dominant signers, this technique can improve recognition rates for playing with either hand. All videos with zero frames of tracking were removed. To deal with variable sequence length, videos with less than 60 frames were padded with a row with values set to -1 at the end of the sample. For videos with more than 60 frames, the middle 60 frames were kept while the rest were discarded. Each of the labels was converted into a one-hot encoding.

| Confused Pair | % Times Confused (% of Dataset) | Confused Pair | % Times confused (% of Dataset) |
|---|---|---|---|
| WAKE, AWAKE | 47.0 (2.7) | PRETEND, HELLO | 15.4 (0.8) |
| GLASSWINDOW, TOOTH | 25.1 (1.2) | NAPKIN, FLOWER | 26.0 (0.8) |
| EYE, TOOTH | 19.8 (1.0) | INTO, ELEPHANT | 11.6 (0.6) |
| CHIN, SAY | 19.9 (1.0) | FINGER, WAIT | 11.4 (0.6) |
| BESIDE, PERSON | 18.8 (0.8) | THAT, STAY | 10.8 (0.4) |

Table 2: Most confused signs

Using Keras [6], the model is built using two bidirectional LSTM layers followed by a dropout layer and the output layer of 250 units with a softmax activation. The first bidirectional LSTM layer has 128 units and has return sequences set to True to allow timestep information to be preserved and passed to the second bidirectional LSTM layer. Default Keras LSTM parameter values are used. The input shape is manually set to (60, 63). The second bidirectional LSTM has 128 units and the default parameter values required for the LSTM layer in Keras remain. The third layer is a dropout layer with rate set to 0.5. The last layer is the 250 unit softmax output layer. During model training, a batch size of 32 is used along with the Adam optimizer and Categorical Cross Entropy Loss. The model is set to train for 40 epochs, and the validation data is used for early callback stopping if the validation loss converges. The weights that results in the lowest validation loss are restored. The model is trained in 30 minutes on one PACE GPU node (4x NVIDIA RTX 6000).

The recognition system achieved 82.1% accuracy (81.9% F1 score) on the verification set and 84.2% accuracy (83.9% F1 score) on the test set for all files that had at least one frame of MediaPipe tracking. When including the files where tracking failed completely (109/29,949 validation; 171/33,667 test), accuracy decreases to 81.9% validation and 83.8% test set, respectively. Upon inspection, the test set has some signers who are highly consistent with their signing, which explains the surprising increase in accuracy between verification and test. The top most confused pair of signs can be seen in Table 2. Many are near homophones mentioned earlier. For example, the video for AWAKE is often labeled WAKE, and vice versa. EYE versus TOOTH confusion is the result of Mediapipe Hands's features not being measured relative to the face. Adding Mediapipe Pose or switching to Mediapipe Holistic, if we can make it run efficiently enough with Unity for PopSign, would significantly improve results.

## 6 Discussion and Future work

With 200,686 total videos, PopSign ASL v1.0 is the largest isolated sign language dataset publicly available. Besides residing at signdata.cc.gatech.edu, we expect to host the 1.1TB PopSign ASL v1.0 dataset at RIT/NTID, UIUC, and DPAN. Unlike previous datasets, PopSign focuses on one-handed signing captured from smartphone selfie cameras. The face and signing hand represent a significant amount of the area of the image. Each video is reviewed manually, with a large majority being labeled as the expected sign. The videos show a large variety of lighting conditions and backgrounds.

The PopSign game only considers five signs at a time. Thus, sets of signs can be chosen to minimize potential confusion by the recognition system. Seven sets of signs were chosen to create PopSignAI Preview, an initial version of the PopSign game with embedded sign recognition that can be found on the Android Play and Apple App stores[7]. For this proof-of-concept a 2D-CNN was trained for each set of five signs and limited to approximately 1MB in size in order to run efficiently on older smartphones such as the Pixel 4A. In informal testing, this limited task increased recognition accuracy to approximately 95% when played by an experienced signer. Gameplay is quite compelling, suggesting that the PopSign ASL v1.0 dataset is suitable for creating sign language recognition based games. Of course, a 250-sign recognition system will allow many more levels of the PopSign game to be created. As the recognition system matures, it will be integrated as a plug-in for game creation engines such as Unity to enable more programmers to create games based on sign language.

---

[6]https://github.com/keras-team/keras

[7]The app is available at https://play.google.com/store/apps/details?id=edu.gatech.popsignai

The lexicon was chosen with a particular application in mind: teaching hearing parents of deaf infants basic sign vocabulary. However, there are approximately 313 concepts in the MacArthur-Bates CDI not covered in the current dataset. A similar collection effort is underway to address these signs, which should expand the usefulness of the data. In speech recognition, small vocabulary isolated recognizers are used for voice dialing ("call" "father"), interactive phone menus ("press one or say 'balance' to get account balances"), multimodal control of windowing systems ("close" while pointing to a window), searching voice mail by voice, and controlling wearable computers ("ok Glass" "show" "compass"). We hope this dataset can support similarly creative use cases for ASL.

Since collection is performed remotely by mailing smartphones to participants, researchers can use our tools and procedures to purposely sample different variants of signing. For example, one can ask participants to intentionally provide different versions of the sign based on their background. Signers who were raised with Black ASL [Valli and Lucas, 2000] or with a local sign language like Hawaiian Sign [Shroyer and Shroyer, 1984] may have used a different set of signs at home than at university. Such a collection allows sign linguists to gain a better understanding of regional signs and enables a general purpose sign recognizer to be more inclusive.

# 7   Ethics and Safety Discussion

"Nothing about us without us" is an often-heard phrase when discussing accessibility with those in the community. Without guidance, such as can be found in the FATE paper Bragg et al. [2021], the many subtleties of sign collection can be confusing. Unfortunately, most machine learning researchers do not know about the existence of the Deaf culture, the fact that ASL has a different grammar than English, or even that there are many distinct sign languages. Here we follow the convention that "deaf" refers to the medical condition, whereas "Deaf" refers to the community, which focuses on sign language. Without Deaf members on the team, researchers can stray into creating technologies that are not useful or are even considered harmful by the community [Erard, 2017, Hill, 2020]. Deaf team members and the Deaf community have informed PopSign at every step. The idea that an educational game might be the first useful application of sign language recognition was posited in 2000 by Dr. Harley Hamilton, a sign linguist and a technology coordinator at the Atlanta Area School for the Deaf. After several iterations of desktop games tested at Deaf schools in Georgia, a student researcher suggested changing the focus to smartphones and introduced the team to the idea that ASL is evolving due to smartphone technology [Morris, 2022a]. PopSign provides the ML community with a clear need and a viable dataset for a goal that can be achieved by sign recognition in the near term. Signers for the dataset were recruited by Deaf researchers and informed before beginning participation about the intent to create a public video database. DPAN's consent procedures were informed by those used by Georgia Tech's IRB. The risks of the dataset videos are similar to any identifiable video made available on the Internet. While names are not affiliated with any of the videos, it was clear in the consent process that participants' faces would be identifiable. Participants were paid $300 for providing 5000 signs, a process that would take approximately six hours. Signers could stop participation without penalty; compensation is prorated. PopSign ASL v1.0 is provided under a Creative Commons CC-BY 4.0 license, specifically so both academics and industry may use it. Software created as part of the collection and review process is covered by the standard MIT license. The dataset is not meant to be representative of ASL, though there is a risk users might use it that way. The next section warns about the dataset's limitations.

# 8   Limitations

While ASL is the most common sign language used in the United States, there are many sign languages both in the United States (e.g., Plains Indian Sign Language and Hawaiian Sign Language) and in the world at large (e.g., British Sign Language, French Sign Language, German Sign Language, Hindi Sign Language, etc.). In addition, there are many regional and cultural accents associated with sign in the United States, including Black American Sign Language. The PopSign dataset is designed for teaching one variant of a signed concept, focusing on one-handed smartphone signing, but it does not capture a representative sample of all the sign variations that would be commonly understood in conversation. ASL has a grammar that is very different from English, and isolated signs do not reflect its richness. A larger number of signers is necessary to better represent skin tones, hand features, ages and different levels of signing fluency. PopSign assumes one-handed sign from

the smartphone viewpoint and field of view, whereas a more general recognition system would also require two-handed signs and a greater variety of viewpoints and camera parameters. As collection required the use of a smartphone, communication via email, and a viable shipping address, it biases the dataset towards younger, more affluent signers with facility in English.

## 9   Conclusion

The Popsign ASL v1.0 dataset provides a large, reviewed dataset of signs with many examples per sign that demonstrate the concepts first taught to infants. It focuses on signing captured by the selfie camera on a smartphone, and the data has already proven viable in creating a sign language recognition based educational game for helping hearing parents of deaf children learn ASL.

## Acknowledgments and Disclosure of Funding

We would like to thank all of the signers who contributed to the project and especially Nathan Qualls and Michaela Jitaru at DPAN for their hard work in data collection. Max Shengelia at RIT helped guide the project toward smartphone signing, based on his observations in the community, and performed initial reviews of the sign. Graham MacKenzie provided perspective as an ASL interpreter on how one-handed smartphone signing is becoming commonplace and provided an initial analysis of which concepts in the CDI could be signed with one hand. RIT/NTID CAT lab is our partner in creating the next PopSign games. Finally, thanks to the Kaggle team (Mark Sherwood, Ashley Chow, and Phil Culliton) for all their advice on how to best create useful datasets. This research was supported in part through research cyberinfrastructure resources and services provided by the Partnership for an Advanced Computing Environment (PACE) at the Georgia Institute of Technollogy, Atlanta, Georgia, USA.

DPAN is a non-profit, and its funding for this project came through non-restrictive gifts by private and public entities. Georgia Tech's contributions are through classwork and volunteer efforts. Google-affiliated personnel play an advisory role and participate on a volunteer basis.

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
