# PopSign ASL v1.0: An Isolated American Sign Language Dataset Collected via Smartphones Supplemental

**Thad Starner**[1]    **Sean Forbes**[2]    **Matthew So**[1]    **David Martin**[1]    **Rohit Sridhar**[1]
**Gururaj Deshpande**[1]    **Sam Sepah**[3]    **Sahir Shahryar**[1]    **Khushi Bhardwaj**[1]
**Tyler Kwok**[1]    **Daksh Sehgal**[1]    **Saad Hassan**[4]    **Bill Neubauer**[1]
**Sofia Anandi Vempala**[1]    **Alec Tan**[1]    **Jocelyn Heath**[1]    **Unnathi Utpal Kumar**[1]
**Priyanka Vijayaraghavan Mosur**[1]    **Tavenner M. Hall**[1]    **Rajandeep Singh**[1]
**Christopher Zhang Cui**[1]    **Glenn Cameron**[3]    **Sohier Dane**[3]    **Garrett Tanzer**[3]
[1]Georgia Institute of Technology    [2]Deaf Professional Arts Network
[3]Google    [4]Rochester Institute of Technology
{thad,matthew.so,dmartin,rohitsridhar,gurudesh}@gatech.edu
{khushi.bhardwaj,tkwok7,dsehgal34,wneubauer4}@gatech.edu
{svempala7,atan70,jheath38,unnathikumar}@gatech.edu
{pmosur3,rajandeep.singh,ccui46}@gatech.edu
sean@dpan.tv, sh253@rit.edu, sahirshahryar@gmail.com
tavennerh@yahoo.com, {sepah, glencam, sohier, gtanzer}@google.com

## 1   Introduction

This supplement to our main paper "PopSign ASL v1.0: An Isolated American Sign Language Dataset Collected via Smartphones" contains the plan for hosting, maintenance, and licensing of the dataset, an author statement affirming of the license, a reproducibility guide for the benchmarks we run on the dataset and a data card describing the dataset.

## 2   Hosting, Licensing, and Maintenance Plan

Our dataset, called PopSign ASL v1.0, will be hosted jointly by the Georgia Institute of Technology and NTID (National Technical Institute for the Deaf). DPAN also plans to host the dataset. It will be available for at least five years, as we have no plans to retire the dataset. The dataset will be available at http://signdata.cc.gatech.edu/, and will be provided under the Creative Commons CC-BY 4.0 license. The data is currently viewable using username: gtccg and password: popsignASL2023 We plan to remove the password protection before the conference. We (the authors) bear all responsibility, including the violation of rights, with respect to the release of this dataset.

## 3   ML Reproducibility

In this section, we describe our model, our experimental setup, the hardware used for training, and the data collection process.

### 3.1   Data Collection

The data collection procedure is given in the data card and the main paper, but we detail it here. The data was collected by DPAN (Deaf Professional Arts Network) using Pixel 4A phones that were shipped to participants. The individual signs videos were recorded in sessions of 10 signs each and

37th Conference on Neural Information Processing Systems (NeurIPS 2023) Track on Datasets and Benchmarks.

uploaded to a Google Photos account. The timestamps corresponding to the sign recordings within these videos were also uploaded and later used to split the videos. Once the videos were split, we used Mediapipe Hands to extract hand tracking features by frame. These features were used for modeling. PACE, a computing cluster at Georgia Tech using the SLURM scheduler to schedule 40 nodes each containing 24 cores to split videos as well as extract Mediapipe Hands features. To create a reproducible environment, a Docker container [1] was used. This Docker container was converted to Apptainer and used on PACE. The scripts for splitting videos and extracting Mediapipe Hands features were each run for 2 hours

After the Mediapipe features were extracted, a script was run to process that data into the same format used for the Kaggle competition. This processing required transforming the .data files created by the Mediapipe Extraction script and formatting them into .parquet files. These parquet files were then placed into train, test, and validation folders based on the users. This task was also run on PACE using a single node with 24 cores for 1 hour.

We preserve eight users for test, eight users for validation and the remaining users for the train set. We further filtered the data using a validation process detailed in the data card. We set up a validation script [2] to quickly validate sign videos.

### 3.2 Model Description

Using Keras [3], a model was built using two bidirectional LSTM layers followed by a dropout layer and the output layer of 250 units with a softmax activation. During model training, a batch size of 256 was used along with the Adam optimizer and Categorical Cross Entropy Loss. The model was set to train for 40 epochs, and the validation data was used for early callback stopping if the validation loss had converged. The weights that resulted in the lowest validation loss were restored. We use PACE for training. The model was trained in 60 minutes on one PACE node (3x NVIDIA RTX 6000) using the interactive shell in Open OnDemand. We set a seed of 11 as well as used the aforementioned Apptainer container to ensure reproducibility.

## 4 Data Details

Table 1 lists the 250 English concepts in alphabetical order with the total number of examples of that concept signed in the manner intended for the "Game." "Total Variants" are the number of examples where the hand motion was recognizable as a sign, but it was not the sign intended to be taught in the game. Also provided are the number of game examples where Mediapipe Hands provided at least one frame of hand landmarks from the video.

Table 2 splits the data set into train, verification, and test sets. The first column is the English concept. The next column indicates how many examples in the training set were the signing intended for the game. The next column is the number of examples in the verification set that matched the signed intended for the game. The "Ver. Var." column indicates the number of examples that were not the sign intended for the game. The final two columns provide similar information for the test set.

| Sign | Total Game | Total Game Tracked by Mediapipe | Total Variants |
|---|---|---|---|
| after | 641 | 638 | 193 |
| airplane | 442 | 431 | 379 |
| all | 495 | 492 | 310 |
| alligator | 813 | 806 | 59 |
| animal | 825 | 811 | 42 |
| another | 851 | 832 | 24 |
| any | 765 | 761 | 98 |
| apple | 710 | 703 | 62 |
| arm | 429 | 415 | 327 |

---

[1] https://hub.docker.com/layers/gurudesh/copycat/copycat-gpu-cuda10.2-cudnn7/images/sha256-91d359f26cf8cfc0f94a4a77d949f3487c03b9c4559ff545f9233a4bba7204a7?context=explore

[2] https://github.com/Benler123/hotkey_annotate

[3] https://github.com/keras-team/keras

| | | | |
|---|---|---|---|
| aunt | 740 | 729 | 63 |
| awake | 808 | 800 | 66 |
| backyard | 487 | 473 | 330 |
| bad | 476 | 462 | 17 |
| balloon | 588 | 585 | 273 |
| bath | 848 | 837 | 19 |
| because | 645 | 626 | 218 |
| bed | 737 | 734 | 55 |
| bedroom | 614 | 603 | 187 |
| bee | 442 | 441 | 336 |
| before | 457 | 443 | 270 |
| beside | 578 | 559 | 235 |
| better | 870 | 860 | 2 |
| bird | 862 | 860 | 2 |
| black | 575 | 570 | 290 |
| blow | 781 | 773 | 89 |
| blue | 735 | 720 | 123 |
| boat | 858 | 836 | 30 |
| book | 859 | 851 | 15 |
| boy | 861 | 848 | 5 |
| brother | 425 | 411 | 450 |
| brown | 829 | 826 | 17 |
| bug | 836 | 835 | 38 |
| bye | 692 | 681 | 188 |
| callonphone | 691 | 680 | 209 |
| can | 834 | 825 | 32 |
| car | 408 | 403 | 59 |
| carrot | 454 | 454 | 327 |
| cat | 782 | 778 | 101 |
| cereal | 457 | 457 | 395 |
| chair | 607 | 593 | 269 |
| cheek | 487 | 478 | 372 |
| child | 708 | 689 | 159 |
| chin | 299 | 299 | 544 |
| chocolate | 820 | 806 | 54 |
| clean | 741 | 734 | 155 |
| close | 423 | 417 | 441 |
| closet | 538 | 538 | 355 |
| cloud | 668 | 667 | 217 |
| clown | 839 | 831 | 37 |
| cow | 758 | 745 | 119 |
| cowboy | 772 | 761 | 67 |
| cry | 613 | 604 | 250 |
| cut | 763 | 744 | 117 |
| cute | 872 | 869 | 5 |
| dad | 868 | 850 | 3 |
| dance | 873 | 853 | 2 |
| dirty | 807 | 805 | 43 |
| dog | 266 | 262 | 617 |
| doll | 868 | 865 | 1 |
| donkey | 795 | 783 | 78 |
| down | 833 | 810 | 24 |
| drawer | 756 | 739 | 117 |
| drink | 862 | 854 | 27 |
| drop | 864 | 849 | 18 |
| dry | 869 | 863 | 13 |
| dryer | 430 | 429 | 426 |
| duck | 756 | 753 | 133 |
| ear | 425 | 422 | 449 |

| | | | |
|---|---|---|---|
| elephant | 716 | 707 | 114 |
| empty | 723 | 711 | 78 |
| every | 328 | 325 | 534 |
| eye | 446 | 439 | 397 |
| face | 778 | 764 | 95 |
| fall | 822 | 807 | 73 |
| farm | 680 | 678 | 207 |
| fast | 536 | 536 | 321 |
| feet | 593 | 592 | 286 |
| find | 398 | 389 | 476 |
| fine | 598 | 595 | 222 |
| finger | 558 | 552 | 278 |
| finish | 792 | 783 | 84 |
| fireman | 649 | 644 | 218 |
| first | 832 | 819 | 49 |
| fish | 510 | 508 | 373 |
| flag | 589 | 585 | 282 |
| flower | 861 | 855 | 23 |
| food | 876 | 869 | 5 |
| for | 865 | 849 | 10 |
| frenchfries | 880 | 870 | 2 |
| frog | 851 | 847 | 11 |
| garbage | 520 | 516 | 361 |
| gift | 849 | 842 | 21 |
| giraffe | 812 | 800 | 52 |
| girl | 780 | 765 | 103 |
| give | 499 | 483 | 351 |
| glasswindow | 484 | 481 | 349 |
| go | 444 | 443 | 410 |
| goose | 579 | 576 | 307 |
| grandma | 754 | 741 | 58 |
| grandpa | 784 | 770 | 35 |
| grass | 853 | 844 | 24 |
| green | 747 | 728 | 124 |
| gum | 858 | 847 | 15 |
| hair | 816 | 809 | 69 |
| happy | 865 | 859 | 5 |
| hat | 770 | 757 | 92 |
| hate | 821 | 808 | 56 |
| have | 864 | 850 | 11 |
| haveto | 792 | 775 | 88 |
| head | 596 | 579 | 299 |
| hear | 492 | 491 | 408 |
| helicopter | 531 | 528 | 306 |
| hello | 860 | 852 | 27 |
| hen | 663 | 657 | 233 |
| hesheit | 752 | 736 | 144 |
| hide | 564 | 560 | 309 |
| high | 874 | 848 | 17 |
| home | 808 | 797 | 2 |
| horse | 875 | 863 | 9 |
| hot | 790 | 779 | 60 |
| hungry | 885 | 876 | 5 |
| icecream | 479 | 474 | 1 |
| if | 695 | 692 | 170 |
| into | 684 | 684 | 154 |
| jacket | 744 | 735 | 120 |
| jeans | 288 | 280 | 544 |
| jump | 743 | 724 | 140 |

| | | | |
|---|---|---|---|
| kiss | 558 | 554 | 285 |
| kitty | 318 | 318 | 547 |
| lamp | 302 | 302 | 548 |
| later | 864 | 852 | 9 |
| like | 830 | 825 | 59 |
| lion | 864 | 857 | 4 |
| lips | 809 | 804 | 83 |
| listen | 396 | 393 | 465 |
| look | 655 | 641 | 238 |
| loud | 627 | 627 | 243 |
| mad | 622 | 612 | 246 |
| make | 874 | 864 | 8 |
| man | 739 | 737 | 146 |
| many | 859 | 851 | 16 |
| milk | 868 | 854 | 8 |
| minemy | 872 | 871 | 10 |
| mitten | 309 | 305 | 539 |
| mom | 843 | 837 | 41 |
| moon | 862 | 853 | 10 |
| morning | 833 | 822 | 48 |
| mouse | 876 | 872 | 26 |
| mouth | 852 | 844 | 25 |
| nap | 253 | 252 | 560 |
| napkin | 514 | 511 | 334 |
| night | 862 | 846 | 33 |
| no | 881 | 869 | 0 |
| noisy | 666 | 655 | 219 |
| nose | 878 | 875 | 3 |
| not | 884 | 868 | 5 |
| now | 857 | 844 | 22 |
| nuts | 603 | 598 | 114 |
| old | 847 | 842 | 36 |
| on | 740 | 728 | 115 |
| open | 802 | 787 | 48 |
| orange | 862 | 855 | 8 |
| outside | 544 | 533 | 144 |
| owie | 693 | 677 | 188 |
| owl | 827 | 822 | 49 |
| pajamas | 672 | 661 | 198 |
| pen | 507 | 502 | 400 |
| pencil | 598 | 592 | 268 |
| penny | 666 | 664 | 215 |
| person | 600 | 591 | 273 |
| pig | 610 | 608 | 273 |
| pizza | 510 | 510 | 352 |
| please | 883 | 878 | 0 |
| police | 167 | 164 | 304 |
| pool | 613 | 607 | 282 |
| potty | 805 | 790 | 67 |
| pretend | 672 | 668 | 185 |
| pretty | 854 | 847 | 0 |
| puppy | 386 | 383 | 479 |
| puzzle | 711 | 701 | 149 |
| quiet | 817 | 803 | 59 |
| radio | 712 | 703 | 169 |
| rain | 871 | 861 | 9 |
| read | 883 | 870 | 11 |
| red | 816 | 809 | 52 |
| refrigerator | 392 | 390 | 459 |

| | | | |
|---|---|---|---|
| ride | 821 | 801 | 53 |
| room | 550 | 547 | 304 |
| sad | 834 | 828 | 45 |
| same | 734 | 734 | 154 |
| say | 699 | 688 | 167 |
| scissors | 888 | 872 | 0 |
| see | 744 | 743 | 142 |
| shhh | 887 | 887 | 1 |
| shirt | 746 | 736 | 99 |
| shoe | 865 | 853 | 9 |
| shower | 96 | 96 | 696 |
| sick | 844 | 835 | 34 |
| sleep | 859 | 854 | 25 |
| sleepy | 197 | 190 | 607 |
| smile | 548 | 540 | 290 |
| snack | 340 | 339 | 483 |
| snow | 687 | 684 | 159 |
| stairs | 738 | 737 | 146 |
| stay | 867 | 863 | 9 |
| sticky | 690 | 670 | 197 |
| store | 789 | 783 | 93 |
| story | 414 | 409 | 432 |
| stuck | 866 | 858 | 4 |
| sun | 784 | 768 | 65 |
| table | 874 | 864 | 6 |
| talk | 802 | 793 | 86 |
| taste | 874 | 860 | 25 |
| thankyou | 864 | 857 | 7 |
| that | 881 | 877 | 4 |
| there | 406 | 393 | 457 |
| think | 260 | 259 | 607 |
| thirsty | 869 | 858 | 6 |
| tiger | 861 | 852 | 12 |
| time | 824 | 811 | 42 |
| tomorrow | 882 | 873 | 1 |
| tongue | 251 | 250 | 613 |
| tooth | 827 | 820 | 62 |
| toothbrush | 690 | 688 | 194 |
| touch | 849 | 844 | 43 |
| toy | 787 | 771 | 95 |
| tree | 885 | 872 | 2 |
| TV | 795 | 783 | 76 |
| uncle | 775 | 773 | 85 |
| underwear | 587 | 587 | 289 |
| up | 835 | 822 | 49 |
| vacuum | 775 | 759 | 89 |
| wait | 844 | 835 | 34 |
| wake | 854 | 846 | 30 |
| water | 636 | 632 | 213 |
| wet | 853 | 840 | 36 |
| weus | 744 | 741 | 132 |
| where | 861 | 844 | 4 |
| white | 836 | 831 | 46 |
| who | 836 | 830 | 46 |
| why | 452 | 452 | 411 |
| will | 805 | 803 | 72 |
| wolf | 879 | 874 | 3 |
| yellow | 830 | 821 | 24 |
| yes | 851 | 840 | 40 |

| | | | | | |
|---|---|---|---|---|---|
| yesterday | 503 | 494 | 385 |
| yourself | 887 | 868 | 1 |
| yucky | 559 | 550 | 328 |
| zebra | 465 | 458 | 33 |
| zipper | 822 | 804 | 52 |

Table 1: List of the English concepts in PopSign, the number of signed examples that match the intended sign for the game, the number of "game sign" examples where Mediapipe provided landmarks, and the number of examples where the sign provided was not the intended sign for the game (i.e., a variant).

| **Sign** | Train Game | Train Var. | Ver. Game | Ver. Var. | Test Game | Test Var. |
|---|---|---|---|---|---|---|
| after | 395 | 156 | 116 | 37 | 130 | 0 |
| airplane | 272 | 270 | 36 | 109 | 134 | 0 |
| all | 291 | 250 | 96 | 60 | 108 | 0 |
| alligator | 534 | 42 | 140 | 17 | 139 | 0 |
| animal | 542 | 41 | 149 | 1 | 134 | 0 |
| another | 557 | 24 | 157 | 0 | 137 | 0 |
| any | 505 | 73 | 129 | 25 | 131 | 0 |
| apple | 479 | 24 | 100 | 38 | 131 | 0 |
| arm | 300 | 187 | 39 | 107 | 90 | 33 |
| aunt | 507 | 13 | 118 | 28 | 115 | 22 |
| awake | 525 | 66 | 157 | 0 | 126 | 0 |
| backyard | 351 | 177 | 37 | 112 | 99 | 41 |
| bad | 307 | 17 | 38 | 0 | 131 | 0 |
| balloon | 426 | 144 | 88 | 69 | 74 | 60 |
| bath | 561 | 18 | 153 | 1 | 134 | 0 |
| because | 450 | 124 | 76 | 82 | 119 | 12 |
| bed | 464 | 47 | 135 | 8 | 138 | 0 |
| bedroom | 397 | 125 | 106 | 44 | 111 | 18 |
| bee | 258 | 234 | 79 | 75 | 105 | 27 |
| before | 289 | 196 | 93 | 51 | 75 | 23 |
| beside | 360 | 197 | 119 | 38 | 99 | 0 |
| better | 579 | 2 | 154 | 0 | 137 | 0 |
| bird | 577 | 2 | 152 | 0 | 133 | 0 |
| black | 401 | 167 | 91 | 68 | 83 | 55 |
| blow | 500 | 73 | 145 | 16 | 136 | 0 |
| blue | 471 | 101 | 133 | 22 | 131 | 0 |
| boat | 563 | 27 | 159 | 3 | 136 | 0 |
| book | 571 | 15 | 153 | 0 | 135 | 0 |
| boy | 574 | 3 | 154 | 2 | 133 | 0 |
| brother | 309 | 278 | 76 | 79 | 40 | 93 |
| brown | 556 | 4 | 143 | 13 | 130 | 0 |
| bug | 544 | 36 | 158 | 2 | 134 | 0 |
| bye | 465 | 123 | 102 | 59 | 125 | 6 |
| callonphone | 476 | 130 | 79 | 79 | 136 | 0 |
| can | 552 | 30 | 152 | 2 | 130 | 0 |
| car | 249 | 58 | 40 | 0 | 119 | 1 |
| carrot | 299 | 256 | 98 | 51 | 57 | 20 |
| cat | 493 | 95 | 156 | 6 | 133 | 0 |
| cereal | 245 | 313 | 103 | 57 | 109 | 25 |
| chair | 407 | 181 | 130 | 25 | 70 | 63 |
| cheek | 342 | 239 | 47 | 106 | 98 | 27 |
| child | 459 | 108 | 132 | 33 | 117 | 18 |

| | | | | | |
|---|---|---|---|---|---|
| chin | 147 | 404 | 25 | 134 | 127 | 6 |
| chocolate | 570 | 16 | 115 | 38 | 135 | 0 |
| clean | 480 | 119 | 125 | 36 | 136 | 0 |
| close | 241 | 333 | 87 | 80 | 95 | 28 |
| closet | 318 | 276 | 85 | 79 | 135 | 0 |
| cloud | 383 | 208 | 149 | 9 | 136 | 0 |
| clown | 567 | 21 | 137 | 16 | 135 | 0 |
| cow | 538 | 49 | 87 | 70 | 133 | 0 |
| cowboy | 500 | 49 | 139 | 18 | 133 | 0 |
| cry | 406 | 163 | 71 | 86 | 136 | 1 |
| cut | 495 | 91 | 132 | 26 | 136 | 0 |
| cute | 580 | 2 | 158 | 3 | 134 | 0 |
| dad | 584 | 2 | 145 | 1 | 139 | 0 |
| dance | 579 | 1 | 156 | 1 | 138 | 0 |
| dirty | 517 | 42 | 151 | 1 | 139 | 0 |
| dog | 153 | 436 | 94 | 64 | 19 | 117 |
| doll | 585 | 0 | 147 | 1 | 136 | 0 |
| donkey | 528 | 51 | 154 | 7 | 113 | 20 |
| down | 545 | 17 | 153 | 7 | 135 | 0 |
| drawer | 498 | 77 | 139 | 22 | 119 | 18 |
| drink | 589 | 8 | 139 | 19 | 134 | 0 |
| drop | 568 | 16 | 159 | 1 | 137 | 1 |
| dry | 572 | 13 | 158 | 0 | 139 | 0 |
| dryer | 228 | 330 | 95 | 67 | 107 | 29 |
| duck | 537 | 66 | 101 | 67 | 118 | 0 |
| ear | 254 | 322 | 77 | 83 | 94 | 44 |
| elephant | 477 | 90 | 128 | 24 | 111 | 0 |
| empty | 465 | 57 | 123 | 21 | 135 | 0 |
| every | 173 | 387 | 54 | 114 | 101 | 33 |
| eye | 252 | 304 | 58 | 93 | 136 | 0 |
| face | 537 | 44 | 123 | 34 | 118 | 17 |
| fall | 540 | 64 | 152 | 7 | 130 | 2 |
| farm | 428 | 164 | 117 | 41 | 135 | 2 |
| fast | 331 | 233 | 93 | 65 | 112 | 23 |
| feet | 400 | 185 | 88 | 72 | 105 | 29 |
| find | 263 | 315 | 98 | 62 | 37 | 99 |
| fine | 335 | 198 | 131 | 23 | 132 | 1 |
| finger | 315 | 240 | 127 | 37 | 116 | 1 |
| finish | 534 | 55 | 121 | 29 | 137 | 0 |
| fireman | 380 | 191 | 131 | 25 | 138 | 2 |
| first | 543 | 47 | 155 | 2 | 134 | 0 |
| fish | 343 | 252 | 72 | 82 | 95 | 39 |
| flag | 395 | 198 | 65 | 84 | 129 | 0 |
| flower | 568 | 23 | 154 | 0 | 139 | 0 |
| food | 593 | 5 | 155 | 0 | 128 | 0 |
| for | 580 | 10 | 152 | 0 | 133 | 0 |
| frenchfries | 592 | 2 | 153 | 0 | 135 | 0 |
| frog | 566 | 10 | 157 | 1 | 128 | 0 |
| garbage | 338 | 247 | 92 | 72 | 90 | 42 |
| gift | 565 | 20 | 152 | 0 | 132 | 1 |
| giraffe | 530 | 50 | 153 | 2 | 129 | 0 |
| girl | 510 | 78 | 132 | 25 | 138 | 0 |
| give | 305 | 271 | 85 | 72 | 109 | 8 |
| glasswindow | 291 | 272 | 81 | 77 | 112 | 0 |
| go | 263 | 310 | 87 | 77 | 94 | 23 |
| goose | 396 | 189 | 77 | 92 | 106 | 26 |
| grandma | 484 | 43 | 137 | 15 | 133 | 0 |
| grandpa | 505 | 23 | 142 | 12 | 137 | 0 |
| grass | 569 | 19 | 148 | 5 | 136 | 0 |

| | | | | | |
|---|---|---|---|---|---|
| green | 469 | 117 | 145 | 7 | 133 | 0 |
| gum | 575 | 12 | 149 | 3 | 134 | 0 |
| hair | 563 | 30 | 118 | 39 | 135 | 0 |
| happy | 571 | 4 | 159 | 1 | 135 | 0 |
| hat | 492 | 83 | 146 | 9 | 132 | 0 |
| hate | 552 | 34 | 150 | 2 | 119 | 20 |
| have | 581 | 10 | 153 | 1 | 130 | 0 |
| haveto | 521 | 71 | 151 | 4 | 120 | 13 |
| head | 378 | 219 | 81 | 80 | 137 | 0 |
| hear | 337 | 256 | 78 | 94 | 77 | 58 |
| helicopter | 393 | 170 | 62 | 96 | 76 | 40 |
| hello | 569 | 24 | 157 | 3 | 134 | 0 |
| hen | 397 | 196 | 131 | 33 | 135 | 4 |
| hesheit | 499 | 99 | 123 | 38 | 130 | 7 |
| hide | 374 | 209 | 58 | 100 | 132 | 0 |
| high | 590 | 13 | 148 | 4 | 136 | 0 |
| home | 525 | 2 | 144 | 0 | 139 | 0 |
| horse | 583 | 8 | 157 | 1 | 135 | 0 |
| hot | 529 | 34 | 132 | 26 | 129 | 0 |
| hungry | 603 | 3 | 150 | 2 | 132 | 0 |
| icecream | 306 | 1 | 40 | 0 | 133 | 0 |
| if | 440 | 126 | 124 | 37 | 131 | 7 |
| into | 454 | 101 | 100 | 53 | 130 | 0 |
| jacket | 490 | 85 | 119 | 35 | 135 | 0 |
| jeans | 186 | 379 | 45 | 126 | 57 | 39 |
| jump | 487 | 109 | 123 | 31 | 133 | 0 |
| kiss | 347 | 196 | 81 | 85 | 130 | 4 |
| kitty | 180 | 406 | 64 | 96 | 74 | 45 |
| lamp | 165 | 397 | 78 | 85 | 59 | 66 |
| later | 574 | 7 | 158 | 2 | 132 | 0 |
| like | 548 | 50 | 151 | 7 | 131 | 2 |
| lion | 577 | 4 | 153 | 0 | 134 | 0 |
| lips | 539 | 59 | 133 | 24 | 137 | 0 |
| listen | 218 | 332 | 76 | 99 | 102 | 34 |
| look | 441 | 157 | 81 | 81 | 133 | 0 |
| loud | 427 | 147 | 82 | 77 | 118 | 19 |
| mad | 440 | 137 | 46 | 109 | 136 | 0 |
| make | 582 | 8 | 155 | 0 | 137 | 0 |
| man | 500 | 90 | 140 | 22 | 99 | 34 |
| many | 566 | 16 | 160 | 0 | 133 | 0 |
| milk | 580 | 7 | 152 | 1 | 136 | 0 |
| minemy | 586 | 10 | 158 | 0 | 128 | 0 |
| mitten | 156 | 432 | 53 | 105 | 100 | 2 |
| mom | 565 | 26 | 147 | 15 | 131 | 0 |
| moon | 578 | 7 | 149 | 3 | 135 | 0 |
| morning | 564 | 25 | 138 | 21 | 131 | 2 |
| mouse | 580 | 25 | 158 | 1 | 138 | 0 |
| mouth | 562 | 17 | 151 | 8 | 139 | 0 |
| nap | 175 | 386 | 37 | 131 | 41 | 43 |
| napkin | 304 | 252 | 78 | 82 | 132 | 0 |
| night | 567 | 32 | 160 | 1 | 135 | 0 |
| no | 591 | 0 | 157 | 0 | 133 | 0 |
| noisy | 488 | 104 | 84 | 75 | 94 | 40 |
| nose | 590 | 3 | 158 | 0 | 130 | 0 |
| not | 590 | 5 | 156 | 0 | 138 | 0 |
| now | 571 | 22 | 150 | 0 | 136 | 0 |
| nuts | 396 | 61 | 67 | 53 | 140 | 0 |
| old | 559 | 35 | 155 | 1 | 133 | 0 |
| on | 497 | 85 | 126 | 28 | 117 | 2 |

| | | | | | |
|---|---|---|---|---|---|
| open | 529 | 47 | 141 | 1 | 132 | 0 |
| orange | 577 | 8 | 152 | 0 | 133 | 0 |
| outside | 339 | 104 | 78 | 40 | 127 | 0 |
| owie | 467 | 125 | 98 | 63 | 128 | 0 |
| owl | 559 | 26 | 134 | 23 | 134 | 0 |
| pajamas | 431 | 157 | 109 | 40 | 132 | 1 |
| pen | 284 | 317 | 83 | 79 | 140 | 4 |
| pencil | 385 | 197 | 111 | 44 | 102 | 27 |
| penny | 400 | 188 | 150 | 4 | 116 | 23 |
| person | 395 | 184 | 89 | 64 | 116 | 25 |
| pig | 329 | 266 | 148 | 4 | 133 | 3 |
| pizza | 313 | 256 | 88 | 74 | 109 | 22 |
| please | 588 | 0 | 159 | 0 | 136 | 0 |
| police | 80 | 217 | 39 | 0 | 48 | 87 |
| pool | 400 | 201 | 78 | 79 | 135 | 2 |
| potty | 547 | 34 | 126 | 33 | 132 | 0 |
| pretend | 431 | 152 | 126 | 33 | 115 | 0 |
| pretty | 578 | 0 | 158 | 0 | 118 | 0 |
| puppy | 247 | 333 | 59 | 100 | 80 | 46 |
| puzzle | 466 | 95 | 112 | 51 | 133 | 3 |
| quiet | 535 | 48 | 147 | 11 | 135 | 0 |
| radio | 482 | 102 | 112 | 47 | 118 | 20 |
| rain | 577 | 8 | 159 | 1 | 135 | 0 |
| read | 588 | 10 | 159 | 1 | 136 | 0 |
| red | 564 | 18 | 135 | 19 | 117 | 15 |
| refrigerator | 241 | 334 | 61 | 100 | 90 | 25 |
| ride | 526 | 53 | 158 | 0 | 137 | 0 |
| room | 344 | 209 | 82 | 83 | 124 | 12 |
| sad | 565 | 25 | 134 | 20 | 135 | 0 |
| same | 484 | 113 | 120 | 35 | 130 | 6 |
| say | 476 | 103 | 99 | 56 | 124 | 8 |
| scissors | 591 | 0 | 158 | 0 | 139 | 0 |
| see | 491 | 108 | 124 | 33 | 129 | 1 |
| shhh | 589 | 0 | 160 | 1 | 138 | 0 |
| shirt | 480 | 78 | 133 | 21 | 133 | 0 |
| shoe | 576 | 9 | 157 | 0 | 132 | 0 |
| shower | 54 | 483 | 23 | 137 | 19 | 76 |
| sick | 562 | 26 | 149 | 8 | 133 | 0 |
| sleep | 569 | 22 | 153 | 3 | 137 | 0 |
| sleepy | 141 | 406 | 17 | 142 | 39 | 59 |
| smile | 360 | 187 | 90 | 70 | 98 | 33 |
| snack | 160 | 377 | 54 | 105 | 126 | 1 |
| snow | 431 | 133 | 129 | 26 | 127 | 0 |
| stairs | 465 | 124 | 137 | 22 | 136 | 0 |
| stay | 574 | 9 | 156 | 0 | 137 | 0 |
| sticky | 475 | 121 | 79 | 75 | 136 | 1 |
| store | 508 | 77 | 147 | 16 | 134 | 0 |
| story | 273 | 290 | 81 | 63 | 60 | 79 |
| stuck | 582 | 3 | 155 | 0 | 129 | 1 |
| sun | 521 | 43 | 130 | 22 | 133 | 0 |
| table | 587 | 6 | 152 | 0 | 135 | 0 |
| talk | 509 | 83 | 154 | 3 | 139 | 0 |
| taste | 583 | 20 | 159 | 1 | 132 | 4 |
| thankyou | 573 | 5 | 152 | 2 | 139 | 0 |
| that | 591 | 4 | 154 | 0 | 136 | 0 |
| there | 318 | 246 | 41 | 120 | 47 | 91 |
| think | 149 | 422 | 34 | 127 | 77 | 58 |
| thirsty | 583 | 6 | 152 | 0 | 134 | 0 |
| tiger | 577 | 12 | 148 | 0 | 136 | 0 |

| | | | | | |
|---|---|---|---|---|---|
| time | 550 | 24 | 136 | 18 | 138 | 0 |
| tomorrow | 591 | 1 | 156 | 0 | 135 | 0 |
| tongue | 158 | 416 | 37 | 119 | 56 | 78 |
| tooth | 553 | 38 | 158 | 3 | 116 | 21 |
| toothbrush | 474 | 107 | 114 | 47 | 102 | 40 |
| touch | 565 | 37 | 150 | 6 | 134 | 0 |
| toy | 513 | 76 | 139 | 19 | 135 | 0 |
| tree | 593 | 2 | 154 | 0 | 138 | 0 |
| TV | 526 | 68 | 147 | 8 | 122 | 0 |
| uncle | 503 | 65 | 139 | 19 | 133 | 1 |
| underwear | 348 | 229 | 100 | 58 | 139 | 2 |
| up | 560 | 32 | 153 | 2 | 122 | 15 |
| vacuum | 507 | 64 | 132 | 25 | 136 | 0 |
| wait | 554 | 33 | 154 | 1 | 136 | 0 |
| wake | 563 | 27 | 153 | 3 | 138 | 0 |
| water | 409 | 157 | 93 | 56 | 134 | 0 |
| wet | 565 | 31 | 157 | 0 | 131 | 5 |
| weus | 457 | 131 | 153 | 1 | 134 | 0 |
| where | 568 | 2 | 155 | 2 | 138 | 0 |
| white | 555 | 25 | 142 | 20 | 139 | 1 |
| who | 545 | 44 | 156 | 2 | 135 | 0 |
| why | 281 | 284 | 108 | 50 | 63 | 77 |
| will | 532 | 53 | 135 | 19 | 138 | 0 |
| wolf | 584 | 3 | 155 | 0 | 140 | 0 |
| yellow | 549 | 21 | 149 | 2 | 132 | 1 |
| yes | 568 | 31 | 144 | 9 | 139 | 0 |
| yesterday | 387 | 200 | 63 | 99 | 53 | 86 |
| yourself | 595 | 1 | 159 | 0 | 133 | 0 |
| yucky | 370 | 218 | 68 | 89 | 121 | 21 |
| zebra | 284 | 32 | 42 | 0 | 139 | 1 |
| zipper | 553 | 30 | 142 | 21 | 127 | 1 |

Table 2: Number of examples of each of the 250 English concepts (with Hands tracking) in the training, verification, and test sets. Examples that were not the intended sign for the game are labeled "variants" and are not used to create the recognition system for PopSignAI.

# 5 PopSign v1.0 Data Card

# PopSign ASL v1.0

95% of deaf children are born to hearing parents. Since many hearing parents do not know sign, these deaf children are at risk for language acquisition delays resulting in cognitive issues. We are making an educational smartphone game PopSign that helps hearing parents practice their signing vocabulary.

Our dataset is the largest collection of isolated sign videos collected using mobile phones. We are using the data to train recognition models for use in smartphone applications, including the PopSign game. PopSign and related educational technology teach hearing parents and deaf children to sign, reducing developmental problems.

**Dataset Link**   https://signdata.cc.gatech.edu

**Data Card Author(s)**

- **Thad Starner, Georgia Tech:** Owner
- **Rohit Sridhar, Georgia Tech:** Contributor
- **Matthew So, Georgia Tech:** Contributor
- **Gururaj Deshpande, Georgia Tech:** Contributor

## Authorship

**Publishers**

**Publishing Organization(s)**

- Georgia Institute of Technology
- Deaf Professional Arts Network

**Industry Type(s)**

- Academic - Tech
- Not-for-profit - Tech

**Contact Detail(s)**

- **Publishing POC:** Thad Starner
- **Affiliation:** Georgia Institute of Technology
- **Contact:** thad@gatech.edu
- **Mailing List:** popsigngame@gmail.com
- **Website:** popsign.org

**Funding Sources**

**Institution(s)**

- Deaf Professional Arts Network

**Funding or Grant Summary(ies)**   DPAN (Deaf Professional Arts Network) is a non-profit. Funding for this project came through non-restrictive gifts to DPAN from both public and private entities.

**Additional Notes:** Georgia Tech contributed to this project through course work and volunteer efforts.

## Dataset Overview

**Data Subject(s)**

- Sensitive Data about people
- Non-Sensitive Data about people

**Dataset Snapshot**

| Category | Data |
|---|---|
| Size of Dataset | 1.1 TB |
| Total Number of Videos | 214,326 |
| Number of Game Videos | 175,022 |
| Total Number of Signs | 250 |
| Total Number of Signers | 47 |
| Average Videos Per Sign | 857 |
| Number of Video Quality Categories | 3 |

**Content Description**   This dataset was collected from October 2022 to March 2023. Videos were sorted into three categories: game (videos which only contained the sign intended to be used for the PopSign game), unrecognizable (videos which clearly did not correspond to any sign and are not included), and variant (videos which contained signs that did not match the game sign). This dataset was collected from October 2022 to March 2023. Videos were sorted into three categories: game (videos which only contained the sign intended to be used for the PopSign game), unrecognizable (videos which clearly did not correspond to any sign and are not included), and variant (videos which contained signs that did not match the game sign). Note that the dataset currently does not contain unrecognizable videos, as these were removed for v1.0 of this dataset. They may be added to future releases.

**Descriptive Statistics**

| Statistic | Game Videos Per Sign | Variant Videos Per Sign |
|---|---|---|
| count | 250 | 250 |
| mean | 700 | 157 |
| std | 179 | 164 |

| Statistic | Game Videos Per Sign | Variant Videos Per Sign |
|-----------|---------------------|-------------------------|
| min | 96 | 0 |
| 25% | 587 | 25 |
| 50% | 764 | 89 |
| 75% | 851 | 272 |
| max | 888 | 696 |

**Sensitivity of Data**

**Sensitivity Type(s)**

- User Content
- User Metadata
- User Activity Data
- Identifiable Data
- S/PII

**Field(s) with Sensitive Data   Intentional Collected Sensitive Data**

(S/PII were collected as a part of the dataset creation process.)

| Field Name | Description |
|------------|-------------|
| Participant Video | Video of participant (upper body captured) |
| Participant Sign | Video of participant performing isolated sign gestures |

**Security and Privacy Handling   Method:** Participants were given a consent form. They were only allowed to record after providing consent for the following:

"The app will collect video and photographic images of Your face, torso, hands, and whatever is in view of the camera(s) along with associated camera metadata (such as color correction, focal length, etc.). … Beyond the video, the following data may be recorded: - The details of each Task, such as the type of Task that was done, signing certain words, or performing specific actions as instructed
- Date and time information associated with the Tasks - Self identified gender - Self identified age range - Self-identified ethnicity - Self assessed sign language proficiency - Signing style information (such as general location where You learned, type of sign learned, age range when you started learning, signing community You are most closely associated with, etc)

As described earlier, if you consent, we will use photos or video clips where your face can be identified. We may use identifiable photos or video clips of you in written or oral presentations about this work and in publicly available on-line databases."

**Risk Type(s)**

- Direct Risk
- Residual Risk

**Risk(s) and Mitigation(s)**   The direct risk involves participants' visual features (their face and body) being linked to their full name. To mitigate this risk, we use anonymized user IDs to identify users. There is still some residual risk. Participants may still be identified using their faces alone. This risk is unavoidable with video data. We have participants sign consent forms acknowledging that they are creating a dataset intended for public use.

**Dataset Version and Maintenance**

**Maintenance Status   Regularly Updated** - New versions of the dataset have been or will continue to be made available.

**Version Details   Current Version:** 1.0

**Release Date:** Sept 2023

**Maintenance Plan   Versioning:** Major updates will be released as a new version, incremented to the nearest tenth from the previous version. For example, if the current version is between 1.0 and 1.09, then a major update will be released as version 1.1. Major updates include the addition of new users and/or new signs. Minor updates are covered below.

**Updates:** If there are missing/extraneous/erroneous videos (error cases described below), any fixes will be released as a new version, incremented by 0.01. E.g. if the current version is 1.0, then any minor updates will be released as 1.01.

**Errors:** Errors in the dataset include incorrectly labeled videos, missing videos, or extraneous videos. Missing videos include videos that the participant recorded but weren't included in the final release. Extraneous videos include videos that only have a partial sign or no sign at all, but were included in the final dataset.

**Feedback:** We will accept feedback via our group email, popsigngame@gmail.com.

**Next Planned Update(s)   Version affected:** 1.0

**Next data update:** TBD

**Next version:** 1.1

**Next version update:** TBD

**Expected Change(s)  Updates to Dataset:** The current dataset includes 250 signs from the MacArthur Bates CDI. We plan to include an additional 313 signs in the new version. It will be recorded by a new set of participants.

## Data Points

**Primary Data Modality**

- Video Data

**Typical Data Point**  A typical data point includes only the full sign motion, with little to no empty space (i.e. no motion) at the beginning or the end of the video. The full sign must be completed within roughly 1 or 2 seconds and a full view of the signing motion must be included. The sign must be the example variant provided in our in house ASL Capture app.

**Atypical Data Point**  An atypical data point may include a lot of empty space (i.e. moments without any motion) at the beginning or end of the video. The full sign may take longer than 1 or 2 seconds to complete. The beginning or the end of the sign may be obscured by poor camera framing, though the sign should still be recognizable. Atypical data points also include signs that are alternate variants or are fingerspelled, rather than the example sign variant provided by the in house ASL Capture App.

## Motivations & Intentions

**Motivations**

**Purpose(s)**

- Research
- Production
- Education

**Domain(s) of Application** `Educational Technology`, `Accessibility`, `Sign Language Recognition`, `Machine Learning`, `Computer Vision`

**Motivating Factor(s)**

- Teaching Sign
- Developing Educational Technology
- Developing Sign Language Recognition

95% of deaf children are born to hearing parents. The communication barrier can cause language deficiencies and cognitive issues in these children. We want to close the gap by developing interactive technologies using sign language recognition, to actively teach hearing parents sign.

Our dataset is collected on mobile phones and is designed to facilitate sign language recognition in mobile games to interactively teach sign. We also hope our example will bring wider focus on the use of interactive machine learning in general to improve educational technology.

**Intended Use**

**Dataset Use(s)**

- Safe for research use

**Suitable Use Case(s)** **Suitable Use Case:** Isolated Sign Language Recognition

**Suitable Use Case:** Isolated Sign Language Recognition for mobile phone applications and games

In general, the data can be used for isolated sign language recognition and related downstream applications.

**Unsuitable Use Case(s)** **Unsuitable Use Case:** Continuous Sign Language Recognition

**Unsuitable Use Case:** Sign to English Translation

The data is not intended for use in continuous sign language recognition, or sign language to English translation.

## Access, Rentention, & Wipeout

**Access**

**Access Type**

- External - Open Access

**Documentation Link(s)**

- Dataset Website URL: https://signdata.cc.gatech.edu

**Retention**

**Duration** The dataset will be available for at least 5 years, but we have no plans to retire the dataset

**Wipeout and Deletion**

**Policy** We do not have plans to retire the dataset, so there is no deletion policy/procedure

## Provenance

### Collection

**Method(s) Used**   We collected data from our mobile recording app, the ASL Capture App. The app presented 10 signs for recording in a single recording session. The entire session was captured on video, but the sign recordings happened during specific time intervals. The participants were presented a sign to record. They then tapped and held a record button to record themselves signing. The timestamps corresponding to the recording intervals for each sign were saved in a separate file.

**Methodology Detail(s)**   **Collection Method:** ASL Capture App

**Platform:** [Platform Name], Google Pixel 4a

**Dates of Collection:** [10 2022 - 02 2023]

**Primary modality of collection data:** - Video Data

**Update Frequency for collected data:** - Static

**Source Description(s)**   Participants were recruited by DPAN

**Collection Cadence**   **Static:** Data was collected once from single or multiple sources.

**Data Processing**   **Collection Method:** ASL Capture App

**Description:** We split session recordings from the ASL Capture App using a python. The resulting split videos were named following this convention: "—.mp4".

**Tools or libraries:** Python, FFMPEG

### Collection Criteria

**Data Selection**

- We selected data based on whether a sign was recognizable or not, i.e. the sign could be clearly made out from the video. We further categorized whether the sign presented was the example sign we (the data collectors) intended for the participant to sign, or another variant.

**Data Inclusion**

- We included any videos that contained a discernible sign and most of the participant's sign was in frame.

**Data Exclusion**

- We excluded any videos that did not contain a full sign from the participant. We also excluded videos where the participant was out of frame so as to make the sign unrecognizable.

## Human and Other Sensitive Attributes

**Sensitive Human Attribute(s)**

- Gender
- Geography
- Language
- Age
- Culture
- Experience or Seniority

**Intentionality   Intentionally Collected Attributes**

Human attributes were labeled or collected as a part of the dataset creation process.

| Field Name | Description |
|---|---|
| Sign Language | Participant's signing style and dialect |
| Gender | Participant's Gender |
| Age Range | Participant's Age |

**Additional Notes:** By providing isolated sign data, participants provided information about their signing style and preferred dialect.

**Unintentionally Collected Attributes**

Human attributes were not explicitly collected as a part of the dataset creation process but can be inferred using additional methods.

| Field Name | Description |
|---|---|
| Geography | Participant's geographic location |
| Culture | Participant's Ethnic Background |
| Seniority | Participant's signing proficiency |

**Additional Notes:** We did not intentionally collect the attributes listed above, but they may be (incorrectly) inferred from the videos. For instance, videos may suggestive of the participant's age or their signing proficiency. Such inferences may be incorrect since may of these attributes cannot be determined using visual cues alone and may depend on the participant's self identification.

**Rationale**  We intended to collect isolated sign language data; hence videos of the participant's signing were collected. The collected attributes (both intentional and unintentional) may be inferred (though not always accurately) from the videos.

**Risk(s) and Mitigation(s)**  The direct risk with this type of video data is with the participant's identity being revealed. For this reason, we use anonymous identifiers. There is still some residual risk with the participant being identified through their faces alone. These participants have signed a consent form (given in the Data Sensitivity section) to address this concern.

## Annotations & Labeling

### Annotation Workforce Type

- Annotation Target in Data

### Annotation Characteristic(s)

| Annotation Type | Number |
|---|---|
| Number of unique annotations | 250 |
| Total number of annotations | 214,326 |
| Average annotations per example | 1 |

**Annotation Description(s)**  **Description:** Annotations were automatically generated with the data, since participants were prompted to record specific signs in each sessions. These sign labels served as the target for the Isolated Sign Language Recognition problem. The annotation count only includes annotations of videos available in the dataset.

### Annotation Distribution(s)

| Sign | Count |
|---|---|
| pen | 907 |
| mouse | 902 |
| call on phone | 900 |
| hear | 900 |
| taste | 899 |

**Above:** We provide the top 5 sign annotations that occur in the dataset. Note that these counts do not include cases where the sign was unrecognizable in the video (these are post-validation counts). To understand our validation procedure, see the next section.

## Validation Types

**Method(s)**

- Annotation/Label Validation

**Description(s)  (Validation Type)**

**Method:** Each video was validated by a team of reviewers. Reviewers would first check whether the video contained a discernible sign and then check whether the sign was the example variant we provided. Only those videos with any discernible sign are kept, while the videos with different sign variants have been preserved for linguistic analysis.

**Platforms, tools, or libraries:**

- Python, openCV

**Description of Human Validators**

**Characteristic(s)  (Validation Type)** - Unique validators: 15 - Number of examples per validator: 14,545 - Training provided: Yes - Expertise required: No

**Description(s)  (Validation Type)**

**Training provided:** We trained validators in watching videos for discernible signs and in detecting the variants. We also trained validators in how to use the tool.

**Validator selection criteria:** We selected validators who had interest in sign language research and who were technically proficient enough to use the Validation tool.

**Gender(s)  (Validation Type)**

- Identifies as Male (36%)
- Identifies as Female (64%)

## Known Applications & Benchmarks

**ML Application(s)**

- Isolated Sign Language Recognition
- Mobile Applications using Sign Language Recognition

**Evaluation Process(es)**  We train an LSTM model designed to output a label (one of the 250 signs) on the training set and then compute accuracies on the validation/test sets.

**Evaluation Result(s)** Evaluation Results - Accuracy (over Total Videos): 82% (Val), 84% (Test)

We provide accuracy in which the denominator consists of all of the originally recorded videos. For a small percent of videos, MediaPipe did not generate features. Note that during PopSign gameplay, players naturally ensure that the hand tracking is showing the overlay skeleton on their hands before signing (i.e., the players are active participants in trying to get the recognition to work). Thus, for the current application, using accuracy measures for the files with Mediapipe Hands features gives a better sense of the accuracy expected during gameplay.