# OpenReview forum: "PopSign ASL v1.0: An Isolated American Sign Language Dataset Collected via Smartphones"
_NeurIPS.cc/2023/Track/Datasets_and_Benchmarks — NeurIPS 2023 Datasets and Benchmarks Poster_

### Official Review · Reviewer_BJjx · 2023-06-28
**Concern about inconsistent labelling**

**Rating:** 7
**Confidence:** 5
**Correctness:** See my concern outlined in "opportuni…

**Strengths:**

This paper addresses an important area of research. I appreciate that there was clearly inclusion of deaf people in the research process, as authors and meaningful contributors. The contributions from DPAN are a particular strength. The specific context of supporting hearing parents in learning ASL is a really excellent one. I commend the authors on choosing an application that is not only interesting from a ML perspective, but is also valuable to the deaf community.

**Additional Feedback:**

None

**Clarity:**

The paper is largely clearly written.

A minor point: The term 'homosigns' is not a technical term. While there was a brief period where some researchers used the term 'cher' to replace 'phon' ('cheremes' instead of 'phonemes' and 'cherology' instead of 'phonology'), this is no longer the case. Having jargon just for sign languages ('homosigns' instead of 'homophones') obscures the commonalities between signed and spoken languages. Using shared terminology for signed and spoken languages reinforces the notion that sign languages are in fact languages.

**Documentation:**

What was the age of ASL exposure and what were the demographic characteristics of the signers (particularly characteristics like race, gender, region, and other individual characteristics that might influence their signing)? It would be important to know how representative the sample might be of the population.

It looks like about 37,000 signs were not rejected or the desired sign. What was the composition of these signs?

**Ethics:**

While I appreciate the need to spell out how damaging language deprivation can be, I urge caution in framing deaf people as being deficient in some way. It is the system for providing language exposure that is the problem, not deaf people. I suggest not referring to language deprivation as a syndrome (i.e., tied to a person) rather than a phenomenon (i.e., tied to an environment), and would remove the term semi-lingual.

I also wish there had been more information guiding users of the dataset as to ethical considerations and sensitivities in the deaf community that they should be aware of. There are a number of potential negative societal impacts of this dataset that were not considered. See this paper for some suggestions.
Bragg, D., Caselli, N., Hochgesang, J. A., Huenerfauth, M., Katz-Hernandez, L., Koller, O., ... & Ladner, R. E. (2021). The fate landscape of sign language ai datasets: An interdisciplinary perspective. ACM Transactions on Accessible Computing (TACCESS), 14(2), 1-45.

**Limitations:**

I am not sure how to interpret the accuracy rate in light of this sentence in the discussion "The PopSign game only considers five signs at a time." Is the accuracy based on the task of choosing the correct match out of five possible options, or is it based on the full set of 250 signs? Much more detail is needed in section 5, and if the accuracy is out of five options that should be explicit in the abstract, as it is not analogous to other ISLR benchmarks.

"Largest publicly available dataset" is true considering the number of examples, but includes a very small vocabulary of only 250 signs.

**Opportunities For Improvement:**

I have three related concerns that completely undermine my enthusiasm for the paper and dataset: the data collection task, the inconsistent labelling, and the purpose of the dataset (isolated sign recognition vs. translation). The data collection procedure is not fully clear in the manuscript, and should have been, but my understanding is that data contributors were shown English glosses of the signs (along with an ASL exemplar video?) and asked to produce and ASL translation. Showing data contributors an English gloss is problem because ASL and English are different languages, and as the authors describe in section 4 each gloss may have multiple translations in ASL. Not only may the English labels correspond to multiple signs (e.g., the English word pretend might correspond to https://www.signingsavvy.com/sign/make+up/7200/1 or https://www.signingsavvy.com/sign/pretend/2190/2), each sign may correspond to multiple labels (e.g., 'hear' and 'listen' could both be reasonable labels for the same sign). The goal of isolated sign recognition technology is to automatically classify all instances of the same sign, and differentiate between instances of different signs. As such, it requires a dataset that has been lemmatized-- labelled to identify all instances of the same sign, and differentiate between all instances of different signs. Lemmatization is a uniquely challenging problem in sign languages, and has been the subject of significant scholarship, none of which was cited here. The lack of consistent labelling (the authors note roughly 20%, but it is not clear where this estimate came from) renders the data of minimal use for the purpose of isolated sign recognition. While the inconsistent labelling could have been corrected in review since the data were all manually checked, lemmatization appears to not have been seriously considered.

Fenlon, J., Cormier, K., & Schembri, A. (2015). Building BSL SignBank: The lemma dilemma revisited. International Journal of Lexicography, 28(2), 169-206.

Schüller, A. (2021). The Lemma Dilemma: Finding relevant lemmas to include in the Communicative Development Inventory for Sign Language of the Netherlands (NGT-CDI).

Hochgesang, J., Crasborn, O. A., & Lillo-Martin, D. (2018). Building the ASL Signbank. Lemmatization Principles for ASL.

The authors acknowledge the inconsistent labelling, but pitch it as a possible strength. This argument falls flat, as it appears to conflate the task of isolated sign recognition with some kind of translation technology (matching an English word to an ASL sign). In section 4 the authors say, "collecting more variations of each sign, which could be valuable in making a more general-purpose sign recognition system." Having more variations of each English word might be useful in a bilingual dataset, but this dataset does not appear to be positioned as a bilingual dataset. First, it is not clear how a lexical level translation corpus (a single English word to a single ASL sign) is useful in general, or specifically for the intended application outlined in section 2. As a bilingual dataset it would be lopsided (includes multiple signs for each English word, and not vice versa). It is not even clear that it would be a useful English -> ASL dataset. For this, I would imagine users should have been encouraged to offer multiple possible translations of each English word. Instead, it appears users were encouraged to use the same translation each time (e.g., the authors say that "it is easy [for participants] to forget which variant was desired"). Because inconsistent labels are used to determine accuracy, it is unclear how to interpret the accuracy metric.

I cannot see a way around this limitation without lemmatizing the data.

**Relation To Prior Work:**

While the information in section 2.0 is more or less correct, the literature cited is extremely old (largely from 15-40 years ago), and is missing a number of seminal works including several by deaf scholars (e.g., Hall, 2017). Work in this space requires interdisciplinary collaboration, and citing the relevant literature in linguistics and deaf studies is a great opportunity to connect the relevant disciplines.

Hall, W. C. (2017). What you don’t know can hurt you: The risk of language deprivation by impairing sign language development in deaf children. Maternal and child health journal, 21(5), 961-965.

Gulati, S. (2018). Language deprivation syndrome. In Language deprivation and deaf mental health (pp. 24-53). Routledge.

Humphries, T., Kushalnagar, P., Mathur, G., Napoli, D. J., Rathmann, C., & Smith, S. (2019). Support for parents of deaf children: Common questions and informed, evidence-based answers. International journal of pediatric otorhinolaryngology, 118, 134-142.

Humphries, T., Kushalnagar, P., Mathur, G., Napoli, D. J., Padden, C., Rathmann, C., & Smith, S. R. (2012). Language acquisition for deaf children: Reducing the harms of zero tolerance to the use of alternative approaches. Harm Reduction Journal, 9(1), 1-9.

Napoli, D. J., Mellon, N. K., Niparko, J. K., Rathmann, C., Mathur, G., Humphries, T., ... & Lantos, J. D. (2015). Should all deaf children learn sign language?. Pediatrics, 136(1), 170-176.

Hall, M. L., Hall, W. C., & Caselli, N. K. (2019). Deaf children need language, not (just) speech. First Language, 39(4), 367-395.

In section 7, Erard, 2017 is a nice citation, but Hill 2020 is by a black deaf author and would be nice to add.

Hill, J. (2020). Do deaf communities actually want sign language gloves?. Nature Electronics, 3(9), 512-513.

Hamilton & Lillo-Martin is from 1986, not 2013.

Berget et al. (2023) is a relevant citation for the point that, "Even “survival level” signing is a worthwhile endeavor for families with deaf children."

Berger, L., Pyers, J., Lieberman, A., & Caselli, N. (2023). Parent American Sign Language skills correlate with child–but not toddler–ASL vocabulary size. Language Acquisition, 1-15.

**Summary And Contributions:**

I was extremely enthusiastic to read this paper, and was optimistic that it would be a tremendous contribution to the field. Unfortunately, the approach to labelling the data appears to be extremely problematic, and I am doubtful that the issue could reasonably be addressed in time for NeurIPS.

---

> ### Author Response · Authors · 2023-08-22
> **Great suggestions: manual review already complete for each signed video for correctness, adding review detail, references, possible negative effects**
>
> Wow!  Thanks for the detailed review! It will definitely improve the paper.
>
> Lemmatization ("labelled to identify all instances of the same sign, and differentiate between all instances of different signs")
>
> Thanks for the references!  We will cite these and add a discussion.  Below, we address the concerns.
>
> Labels are consistent:
>
> Each video is manually reviewed as to assure that it is the same sign as shown in the example video. On signdata.cc.gatech.edu, videos in archives labeled game-<SIGN>.tar are the same sign as shown in the example video (165,198 total). To be clear, it is not a sign that could possibly be glossed as the same English word; it is the *same* sign and would be recognized as such by a Deaf signer. (To be precise, different numbers of repetitive motions are allowed when the sign is clearly the same and the meaning is not changed.  For example, SHOWER is recognizable whether the motion is repeated 1x, 2x, or 3x. ASL Citizen takes the same approach: the motion for SHOWER is most often done once, but 2x and 3x are included in the 30 examples of SHOWER in that dataset.)
>
> Videos in archives labeled variant-<SIGN>.tar are signs that are different from the one in the example video (35,488 = 17.5% - instead of the 20% stated in the original draft). Table 1 in the supplementary material provides a breakdown for each sign.  We expect most users of the dataset to ignore these archives, but we provide it for completeness and believe it could be useful in the future (perhaps predicting what properties of a gloss or sign might most lead to variations). In future work, the variant-<SIGN>.tar archives will be labelled as to which sign(s) are represented in the archive. The rest of the videos were rejected as not containing a sign, having other identifiable people in the background, or another issue making it unusable.  We will improve the writing to clarify these points.
>
> Choice of signs to collect:
>
> The original set of CDI example sign videos were produced by sign linguist Dr. Harley Hamilton as part of the SMARTSignDictionary 20 years ago. That dictionary goes to significant pains to separate English words that can have several signed meanings ("run") and signs that can have several glosses (OF/BELONG). However, as we are interested in one-handed versions of the CDI signs to teach hearing parents rudimentary vocabulary on their smartphones, collaborator DPAN re-shot every example video, sometimes observing that a SMARTSignDictionary sign was "old" or "regional." A team of five Deaf signers at DPAN (signing since childhood) consulted to determine how each sign should be formed, assuming they were signing one-handed on a mobile phone video conference and reviewing issues such as regional signing versus more common signs (DPAN is in Detroit, but the personnel come from schools in many regions, including Rochester). In the few cases where there was no adequate way to produce a sign one-handed and the team agreed that they would fingerspell the sign if on a one-handed video call, the fingerspelling was provided as the example.  For the first 250 CDI signs (out of the 562 we intended to collect long term), we tried to avoid collecting homophones and near homophones, though some still remained (see Table 2).  We were not aware of Caselli's 2020 CDI 2.0, but we can leverage it in our future work.
>
> Data collection task:
>
> Yes, participants were prompted by English words.  Clicking on that word showed a video of the sign intended to be collected, and participants were instructed by DPAN to use that video for reference. Even so, unintended signs were collected, but they were separated from the intended signs in the manual review process. The collection app has been updated to show a thumbnail video of the sign to be collected directly on the collection screen.
>
> Accuracy rate
>
> The accuracy is based on selecting which of 250 labels a given video should be assigned.  In other words, random chance would be 0.4% and our current accuracy is 85%. We will make that clearer in the text.  If possible, we will add results of 100 rounds of testing the recognizer on 5 random signs chosen from the 250 to give an estimation of gameplay recognition. Given current gameplay, we expect it to be in the high 90s.
>
> Homosigns -> homophones
>
> We will make this change
>
> References
>
> THANK YOU! Some are new to us. We will work/re-add these into the revision.
>
> Demographics & Purpose
>
> We do not have age of learning sign, but all participants self-reported that ASL was their primary language. We can add the demographics we have (it is in our consent form).  Our effort was to collect examples of specific signs to teach hearing parents basic vocabulary, not variations that depend on race and region.
>
> Ethics
>
> We will rephrase the language deprivation/semi-lingual section as suggested. We are using Bragg's FATE paper to add info on potential negative societal impacts and guiding users of the dataset as to ethical concerns and sensitivities.

---

> > ### Comment · Reviewer_BJjx · 2023-08-28
> >
> > These clarifications are very helpful, and the answers about the labelling process in particular substantially increase my view of the paper.

---

> > > ### Author Response · Authors · 2023-08-29
> > > **Ethics - please help complete!**
> > >
> > > Reviewer BJjx-  If you get a chance, can you comment on other specific ethics issues we should cover?  Anything we are missing from the paper and revisions below?
> > >
> > > The main danger we see is the folks who do not know ASL or the Deaf culture will consider this dataset representative of sign.  We plan  to make the limitations and biases much more explicit, talking about
> > >
> > > * it is one-handed sign (!!!)
> > > * there are many signed languages - even in the US
> > > * many regional variations
> > > * it is isolated - i.e., co-articulation issues, classifier grammars, spatial references, etc. are not represented
> > > * it focuses on one way to sign a concept - there can be many
> > > ** talk about how signs were chosen more: CDI + 5 signers debating
> > > * while attempted to get skin tones and gender balance, more signers are needed
> > > * selection bias for people who have a mailing address, have wifi, can use a mobile phone, have email (implies income, some English facility)
> > > * bias towards people near Detroit
> > >
> > > and more technical things like
> > > * compression
> > > * encouraged varied background
> > > * limited use cases (see response to other reviewer about uses of limited one-handed sign)
> > >
> > > What else?  (I am not clear if we are supposed to submit a revision before end of Tuesday with the extra page that is allowed.)

---

> > > > ### Author Response · Authors · 2023-08-31
> > > > **New PDF submitted**
> > > >
> > > > New PDF submitted based on the suggestions here

---

### Official Review · Reviewer_cDoQ · 2023-07-05
**The PopSign ASL v1.0 dataset is a large, reviewed, public dataset containing one handed signs.**

**Rating:** 7
**Confidence:** 4
**Clarity:** The paper is well-written and effecti…

**Strengths:**

The introduced dataset comprises a remarkable number of sign videos compared to other ASL datasets, and the review process for these videos is comprehensive.

The designed app is user-friendly and incorporates gamification, which could expedite parents' learning of sign language. During sign practice, the selfie camera is activated, enabling real-time hand recognition of the user. Additionally, rendered line approximations overlaid on the signing hand aid the user.

**Additional Feedback:**

No further comments.

**Correctness:**

The construction of the dataset is logical and mostly balanced.

The authors used a bidirectional LSTM, which is a reasonable choice for video classification. They only published the validation and test accuracy values, other metrics, such as F1 score, are not mentioned.

**Documentation:**

Detailed description of the dataset is available in the supplemental material.

**Ethics:**

The dataset comprises videos of participants performing signs, where their faces are also identifiable. The participants have given consent for the publication of their data. While the videos do not include names associated with the recordings, the faces of the participants can be recognized.

**Limitations:**

The limitations are addressed, such as the different accents of sign language.

**Opportunities For Improvement:**

The dataset contains videos of the signers switching their signing hands, but the left handed signs were flipped to convert them into right-handed signs. By having signs performed with both hands would help the generalizability of the models and the app users to practice the signs with both hands.

The inclusion of a dictionary feature in the app would be beneficial, allowing users to review previously learned signs at their convenience.

**Relation To Prior Work:**

The PopSign game is  built on past user studies, which is referred.

The authors compared their introduced dataset to 6 previous ASL and 15 other sign language datasets.

**Summary And Contributions:**

This paper introduces the largest publicly available dataset of isolated signs. The dataset consists of videos featuring one-handed signs commonly used during video calls on smartphones. It includes recordings of 250 isolated ASL signs captured using smartphone selfie cameras.

The authors have published the PopSign smartphone app, designed to assist hearing parents of deaf infants in learning sign language to teach their children. The app features a user-friendly game design that not only displays signs and their meanings but also enables users to practice the signs.

---

> ### Author Response · Authors · 2023-08-22
> **F1 score, hand switching, and dictionary**
>
> Thanks for the suggestions and the positive review!
>
> 1. F1 score
>
> We will add the F1 score to the paper. Should we add other metrics as well? Current results are
>
> Test Accuracy: 0.8511922528438298
> Val Accuracy: 0.826038873994638
> Test F1 Score: 0.8481284086237323
> Val F1 Score: 0.8248407525317218
>
> 2. adding dictionary for players
>
> Thank you for the suggestion. We added this feature to the original PopSign game (without recognition) but had to remove it to reduce the complexity of PopSignAI (with recognition) and reduce its size so that it would fit inside the size guidelines on the iPhone App store.  We hope to add it back as the game is developed.
>
> 3. left handed signing
>
> PopSignAI can be played right or left handed.  When the PopSignAI game sees that the signer is playing left-handed, it automatically flips the input to the recognition system. In current informal user testing, accuracy is good for both left and right-handed players, and both enjoy the gameplay.
>
> 4. Ethics
>
> Thanks for flagging for ethics review. We were happy to have more experts looking at what we did. It seems the first ethics reviewer saw no problems and reviewer 2 suggested following up on the suggestions by reviewer Bjjx (see the response to that review for more detail).  We will do so!

---

> > ### Comment · Area_Chair_pEqF · 2023-08-29
> > **Reviewer cDoQ, pls check the rebuttal and see whether you need update the comments and rating?**
> >
> > Reviewer cDoQ, pls check the rebuttal and see whether you need update the comments and rating?

---

### Official Review · Reviewer_HKTi · 2023-07-21
**Reviews of HKTi**

**Rating:** 6
**Confidence:** 4
**Correctness:** Yes. The claims are correct.
**Clarity:** Yes. It is well written.

**Strengths:**

1. It is interesting to enable sign recognition for this kind of educational application. It may help to attract more users.
2. It consists of over 200K videos with a high resolution, which is the largest one to the best of my knowledge.
3. The videos are manually reviewed leading to high quality.

**Additional Feedback:**

N/A

**Documentation:**

Yes. The authors provide sufficient documentation.

**Ethics:**

No.

**Limitations:**

Yes. The authors have discussed limitations.

**Opportunities For Improvement:**

1. The current form of the dataset is mainly used for the recognizer of PopSign. It is not clear to me how it can boost the research on sign language understanding given its limited vocabulary size.
2. The dataset only supports single-handed signs, which limits the expressiveness of signs.
3. Is there any human feedback over the PopSign equipped with the sign recognizer? Since there must be incorrect predictions of the recognizer, is the accuracy of the current system (77% on the validation set) good enough for the realistic usage?
4. More details for the recognition model are needed. For example, the authors may add more details about the inputs of the model (videos or keypoints) and feature extraction (I only see the authors use LSTM layers. Are any other layers like CNNs used to process videos?).

**Relation To Prior Work:**

Yes. A detailed comparison is in Table 1.

**Summary And Contributions:**

The paper introduces an isolated American sign language (ASL) dataset, PopSign ASL v1.0. It aims to enable sign recognition for an ASL educational mobile application, PopSign, which helps parents whose infants are deaf to learn ASL when playing a game. The dataset is the largest one in terms of the video number among ASL datasets.

---

> ### Author Response · Authors · 2023-08-22
> **Adding details on model, usefulness, and limitations**
>
> Thanks for the review!
>
> 1) small dictionary
>
> We agree that creating an isolated dataset with a small vocabulary is limiting. However, we believe the PopSignAI game (now available on iOS and Android) is an important application for the Deaf community and is one of the few situations where current recognition is accurate enough to be both useful and usable right now. Already we are seeing hearing parents of deaf infants use the game, as well as people who just want to learn some ASL. However, sign language recognition is still in very early stages.  Even early versions of DragonDictate in 1992, which had limited vocabulary isolated speech recognition for text entry, was more practical than most current sign recognition systems. However, innovators like Chris Schmandt at the MIT Media Laboratory showed use cases for isolated speech recognition (see his "Voice Communication with Computers" book) such as voice dialing ("call" "father"), interactive phone menus ( "press one or say balance to get account balances"), multimodal control of windowing systems ("close"  while pointing to a window) , the famous "Put That There" demo that combined gesture with speech for a command system, and searching voice mail by voice.  Even in 2014, the Google Glass head worn display (and in-car speech systems of that period) had many short commands that were recognized on device and could be chained together for useful functionality ("OK GLASS" "SHOW" "COMPASS"). While a 250 sign dictionary is small, we hope the community can make creative use of it, much as did speech interface creators from 30 years ago.  Perhaps it would be useful to add some of this motivation to the paper?
>
> 2. one-handed
>
> Yes, we agree one-handed sign can limit expressiveness, and we mostly focus on signs that are naturally one-handed for the PopSign game.  However, the signers and interpreters in our team remark at how one-handed signing is becoming commonplace due to mobile video conferencing. We believe one handed "smartphone" sign is an opportunity for linguists to study the living, evolving nature of sign, and we will be revisiting this idea in our future studies and datasets.
>
> 3. human feedback on recognition accuracy
>
> The PopSign game needs only to distinguish one of five signs for each level of the game.  Thus, our current 85% accuracy on 250 signs can reach the high 90s in actual gameplay.  If possible, we will add results of 100 rounds of testing the recognizer on 5 random signs chosen from the 250 to give an estimation of gameplay recognition. While structured user testing of the game will be the subject of future publications, the game is most assuredly fun to play currently (search for PopSignAI on Android or iOS).
>
> 4. more details on recognition
>
> We will add more details on the recognition system and have folks in our department see if they understand it sufficiently to reproduce it.  Currently, the architecture is pretty straightforward.  Video processing is done by Google's MediaPipe, which generates hand landmarks for the LSTM.

---

> > ### Comment · Reviewer_HKTi · 2023-08-29
> >
> > Thanks for the authors' rebuttal. I believe that this dataset is impactful on sign language recognition for educational needs. I still lean to accept this paper.

---

### Author Response · Authors · 2023-08-22
**Are we supposed to submit a revision by Aug 29th?**

Folks-

Are we supposed to submit a revision by Aug 29th based on the reviewers' feedback?  And if so, can we add the extra page?

Thad

---

> ### Author Response · Authors · 2023-08-31
> **submitted**
>
> a revision has been submitted, which includes the 10th page that can be used to help address reviewers comments

---

> > ### Author Response · Authors · 2023-08-31
> > **final revision update added**
> >
> > Did a final revision on the main text and to update the supplemental information and datacard

---

### Decision · Program_Chairs · 2023-09-22

**Decision:**

Accept (Poster)

**Comment:**

The paper introduces an isolated American sign language (ASL) dataset, PopSign ASL v1.0. It aims to enable sign recognition for an ASL educational mobile application, PopSign, which helps parents whose infants are deaf to learn ASL when playing a game. The dataset is the largest one in terms of the video number among ASL datasets.

The Ethics Reviews are generally positive.

One reviewer has concerns on dictionary size, one-handed and recognition accuracy, the rebuttal is sucessful and the reviewer leans to accept the paper though the score is 6. The other two reviewers are positive and one reviewer is satisfied with the rebuttal.

Based on the comments, rebuttals and discussions, this work is recommended for accept as poster.